# IMP-MARL: a Suite of Environments for Large-scale Infrastructure Management Planning via MARL

**\*Pascal Leroy**[1♯], **\*Pablo G. Morato**[2♭], **Jonathan Pisane**[3], **Athanasios Kolios**[2], **Damien Ernst**[1,4]

[1]Montefiore Institute, University of Liège, [2]Technical University of Denmark, [3]Thales Belgium
[4]LTCI, Telecom Paris, Institut Polytechnique de Paris

[♯]`pleroy@uliege.be`, [♭]`pgmdo@dtu.dk`
[*]Authors contributed equally

## Abstract

We introduce IMP-MARL, an open-source suite of multi-agent reinforcement learning (MARL) environments for large-scale Infrastructure Management Planning (IMP), offering a platform for benchmarking the scalability of cooperative MARL methods in real-world engineering applications. In IMP, a multi-component engineering system is subject to a risk of failure due to its components' damage condition. Specifically, each agent plans inspections and repairs for a specific system component, aiming to minimise maintenance costs while cooperating to minimise system failure risk. With IMP-MARL, we release several environments including one related to offshore wind structural systems, in an effort to meet today's needs to improve management strategies to support sustainable and reliable energy systems. Supported by IMP practical engineering environments featuring up to 100 agents, we conduct a benchmark campaign, where the scalability and performance of state-of-the-art cooperative MARL methods are compared against expert-based heuristic policies. The results reveal that centralised training with decentralised execution methods scale better with the number of agents than fully centralised or decentralised RL approaches, while also outperforming expert-based heuristic policies in most IMP environments. Based on our findings, we additionally outline remaining cooperation and scalability challenges that future MARL methods should still address. Through IMP-MARL, we encourage the implementation of new environments and the further development of MARL methods.

## 1 Introduction

Intelligent agents trained with reinforcement learning (RL) have proven successful in solving complex decision-making tasks, e.g., games [1–3], autonomous driving [4, 5], human healthcare [6], nuclear fusion [7], among others. RL training approaches where multiple agents interact together are commonly denoted as multi-agent reinforcement learning (MARL) methods. In certain applications, these agents must cooperate to accomplish a common goal, leading to the special case of cooperative MARL. To support the advancement of cooperative MARL methods, multiple environments based on games and simulators have served as benchmark testbeds, e.g., the particle environment [8], StarCraft Multi-Agent Challenge (SMAC) [9, 10], and MaMuJoCo [11]. Benchmarking on environments based on games and simulators is useful for the development of MARL methods in specific collaborative/competitive tasks, but additional challenges may still be encountered when deploying MARL methods in real-world applications [12].

Infrastructure Management Planning (IMP) is a contemporary application that responds to current societal and environmental concerns. In IMP, inspections, repairs, and/or retrofits should be timely planned in order to control the risk of potential system failures, e.g., bridge and wind turbine failures,

37th Conference on Neural Information Processing Systems (NeurIPS 2023) Track on Datasets and Benchmarks.

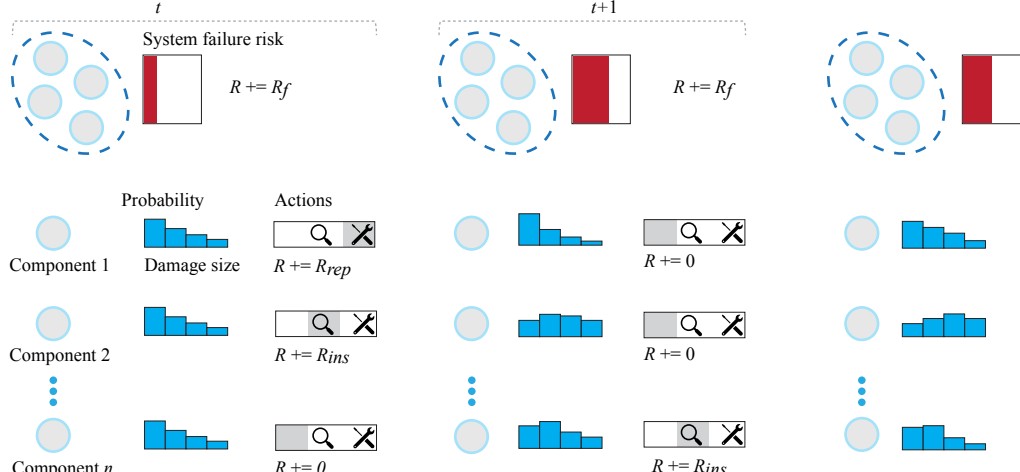

Figure 1: Overarching representation of an infrastructure management planning (IMP) problem. The system failure risk is defined as a function of the probability distribution over the components' damage condition. To control the system failure risk, components can be inspected or repaired at each time step $t$ and, typically, an agent controls one component. The objective of IMP's problem is the maximisation of the expected sum of discounted rewards, by balancing the system failure risk $R_f$ against inspections $R_{ins}$ and repairs $R_{rep}$, all three being negative rewards. Here, we show three components with the same damage probability at time step $t$. When a component is not inspected nor repaired, its damage probability evolves according to a deterioration process. If a component is inspected, information from the inspection is also considered when updating the damage probability, and if a component is repaired, the damage probability resets to its initial damage distribution.

among many others [13]. Formally, the system failure risk is defined as the system failure probability multiplied by the consequences associated with a failure event, typically defined in monetary units. Due to model and measurement uncertainties, the components' damage is not perfectly known, and decisions are made based on a probability distribution over the damage condition, henceforth denoted as damage probability. The system failure probability is defined as a function of components' damage probabilities. Starting from its initial damage distribution, each component's damage probability transitions according to a deterioration stochastic process, but also according to the decisions made [13]. Naturally, the damage probability transitions based on its deterioration model when the component is neither inspected nor repaired, i.e., do-nothing action. If a component is inspected, its damage probability is updated with respect to the inspection outcome. When a component is repaired, its damage condition is directly improved and the damage probability resets to its initial damage distribution. A schematic of a typical IMP problem is shown in Figure 1.

In an effort to generate more efficient strategies for managing engineering systems through the application of cooperative MARL methods, we introduce IMP-MARL, a novel open-source suite of multi-agent environments. In IMP-MARL, each agent is responsible for managing one constituent component in a system, making decisions based on the damage probability of the component. Besides seeking to reduce component inspection and maintenance costs, agents should effectively cooperate to minimise the system failure risk. With IMP-MARL, our goal is to facilitate the definition and implementation of new customisable environments. By jointly minimising system failure risks and inspection/maintenance costs, more effective IMP policies contribute to a better allocation of resources from a societal perspective. Furthermore, additional societal impact is also made by controlling the risk of system failure events. For example, the failure of a wind turbine may affect the available electricity production. Beyond economic considerations, our proposed IMP-MARL framework can also be used to include sustainability and societal metrics within the objective function by accounting for those directly in the reward model.

To assess the capability of cooperative MARL methods for generating effective policies for IMP problems involving many components, we additionally benchmark here state-of-the-art cooperative MARL methods in terms of scalability and optimality. Most of the benchmarked methods are centralised training with decentralised execution (CTDE) methods [14, 15], in which each agent

acts based on only local information, while global information can be utilised during training. Specifically, we benchmark five CTDE methods: QMIX [16], QVMix [17], QPLEX [18], COMA [19], and FACMAC [11], along with a decentralised method, i.e., IQL [20], and a centralised one, i.e., DQN [1]. All tested MARL methods are compared against expert-based heuristic policies, which can be categorised as a state-of-the-art method to deal with IMP problems in the reliability engineering community [13, 21]. In our study, three sets of IMP environments are investigated, including one related to offshore wind structural systems, where MARL methods are tested with up to 100 agents. Additionally, these environments can be set up with two distinct reward models, and one of them incorporates explicit cooperative objectives. For the sake of enabling the reproduction of any published result, we have made our best effort to ensure that the necessary code is publicly available.

Our contributions can be outlined as follows:

- We introduce IMP-MARL, a novel open-source suite of environments, motivating the development of scalable MARL methods as well as the creation of new IMP environments, enabling the effective management of multi-component engineering systems and, as such, leading to a positive societal impact.
- In an extensive benchmark campaign, we test state-of-the-art cooperative MARL methods in very high-dimensional IMP environments featuring up to 100 agents. The resulting management strategies are evaluated against expert-based heuristic policies. We publicly provide the source code for reproducing our reported results and for easing direct comparisons with future developments.
- Based on our results, we draw relevant insights for both machine learning and reliability engineering communities, further highlighting important challenges that must still be resolved. While cooperative MARL methods can learn superior strategies compared to expert-based heuristic policies, the relative performance benefit decreases in environments with over 50 agents. In certain environments, cooperative MARL policies are characterised by a high variance and sometimes underperform expert-based heuristic policies, suggesting the need for further research efforts.

## 2   Related work

**MARL environments** Cooperative MARL has a long-standing history, and decentralised approaches such as IQL were already originally proposed in 1993 [20]. There has been a recent interest in the development of CTDE methods [14, 15] (see Section 4.1), inducing the creation of new environments with cooperative tasks. With continuous action spaces, popular environments include the particle environment [8], a suite of communication oriented environments with cooperative scenarios, and MaMuJoCo [11], which aims at factorising the decision of MuJoCo [22], a physics-based simulator. In contrast, the StarCraft multi-agent challenge (SMAC) [9] and its upgraded version SMACv2 [10] are probably the most studied environments with discrete action spaces. SMAC is based on the StarCraft II Learning Environment [23] with a suite of micro-management challenges where each game unit is an independent agent. Other cooperative environments based on game simulators include the Hanabi Challenge [24], a "cooperative solitaire" between two and five players, and Google Research Football [25], a football game simulator. Cooperative MARL methods are mostly benchmarked on these games and simulators, but also on real-world applications: target coverage control (MATE) [26], train scheduling (Flatland-RL) [27], traffic control (CityFlow) [28], multi-robot warehouse (RWARE) [29][30]. Oroojlooy and Hajinezhad [12] provide a review of cooperative MARL, including a more detailed list of applications. IMP-MARL introduces two key advancements: support for environments with up to 100 agents and seamless creation of diverse IMP environments. This enables the utilisation of RL in real-world scenarios, ranging from complex factories with heterogeneous components to offshore wind farms with multiple homogeneous components.

**Infrastructure management planning methods** Recent heuristic-based inspection and maintenance (I&M) planning methods generate IMP policies based on an optimised set of predefined decision rules [21, 31]. By evaluating only a set of decision rules out of the entire policy space, the previously mentioned approaches might yield suboptimal policies [13]. In the literature, one can also find POMDP-based methods applied to the I&M planning of engineering components, in most cases, relying on efficient point-based solvers [13, 32, 33]. When dealing with multi-component engineering systems, solving point-based POMDPs becomes computationally complex. In that case, the policy and value function can be approximated by neural networks, enabling the treatment of high-dimensional engineering systems. Specifically, actor-critic, and value function-based methods have been proposed in the literature for the management of engineering systems [34–36] with some of them relying on

CTDE methods [37, 38]. Note that no open-source methods nor publicly available environments are provided in the above-mentioned references. This emphasises the importance of our efforts to enhance comparison and reproducibility within the reliability engineering community.

## 3 IMP-MARL: A suite of Infrastructure Management Planning environments

In IMP, the damage condition of multiple components deteriorates stochastically over time, inducing a system failure risk that is penalised at each time step. To control the system failure risk, components can be inspected or repaired, yet, incurring additional costs. The objective is the minimisation of the expected sum of discounted costs, including inspections, repairs, and system failure risk. This can be achieved through the agents' cooperative behaviour, assigning component inspections and repairs while jointly controlling the system failure risk. The introduced IMP decision-making problem can be modelled as a decentralised partially observable Markov decision process (Dec-POMDP).

### 3.1 Preliminaries

A Dec-POMDP [39] can be defined by a tuple $[\mathcal{S}, \mathcal{Z}, \mathcal{U}, n, O, R, P, \gamma]$, where $n$ agents simultaneously choose an action at every time step $t$. The state of the environment is $s_t \in \mathcal{S}$ where $\mathcal{S}$ is the set of states. The observation function $O : \mathcal{S} \times \{1, .., n\} \rightarrow \mathcal{Z}$ maps the state to an observation $o_t^a \in \mathcal{Z}$ perceived by agent $a$ at time $t$, where $\mathcal{Z}$ is the observation space. Each agent $a \in \{1, .., n\}$ selects an action $u_t^a \in \mathcal{U}_a$, and the joint action space is $\mathcal{U} = \mathcal{U}_1 \times .. \times \mathcal{U}_n$. After the joint action $\boldsymbol{u_t} \in \mathcal{U}$ is executed, the transition function determines the new state with probability $P(s_{t+1}|s_t, \boldsymbol{u_t}) : \mathcal{S}^2 \times \mathcal{U} \rightarrow \mathbb{R}^+$, and $r_t = R(s_{t+1}, s_t, \boldsymbol{u_t}) : \mathcal{S}^2 \times \mathcal{U} \rightarrow \mathbb{R}$ is the team reward obtained by all agents. An agent's policy is a function $\pi^a(u_t^a|\tau_t^a, o_t^a) : (\mathcal{Z} \times \mathcal{U}_a)^t \rightarrow \mathbb{R}^+$, which maps its history $\tau_t^a \in (\mathcal{Z} \times \mathcal{U}_a)^{t-1}$ and its observation $o_t^a$ to the probability of taking action $u_t^a$. The joint policy is denoted by $\boldsymbol{\pi} = (\pi^1, .., \pi^n)$. The cumulative discounted reward obtained from time step $t$ over the next $T$ time steps is defined by $R_t = \sum_{k=0}^{T-1} \gamma^k r_{t+k}$ and $\gamma \in [0, 1)$ is the discount factor. The goal of agents is to find the optimal joint policy that maximises the expected $R_t$ during the entire episode: $\boldsymbol{\pi^*} = \text{argmax}_{\boldsymbol{\pi}} \mathbb{E}[R_0|\boldsymbol{\pi}]$.

### 3.2 Environments formulation

**States and observations** As introduced, each agent in IMP perceives $o_t^a$, an observation corresponding to its respective component damage probability and the current time step. Each component damage probability transitions based on a deterioration model, defined according to physics-based engineering models, e.g., numerical simulators and/or analytical laws [32]. The damage probability is also updated based on maintenance decisions, as explained in Section 1. Since the components' damage is not perfectly known, the state of the Dec-POMDP is defined as the collection of all components' damage probabilities along with the current time step: $s_t = (o_t^1, .., o_t^n, t)$.

**Actions and rewards** Each agent controls a component and collaborates with other agents in order to minimise the system failure risk while minimising local costs associated with individual repair and/or inspection actions. At each time step $t$, an agent decides $u_t^a$ between (i) do-nothing, (ii) inspect, or (iii) repair actions, as described in Section 1. Both inspection and repair actions incur significant costs, formally included in the Dec-POMDP framework as negative rewards, $R_{ins}$ and $R_{rep}$, respectively. Moreover, the system failure risk is defined as $R_f = c_F \cdot p_{F_{sys}}$ where $p_{F_{sys}}$ is the system failure probability and $c_F$ is the associated consequences of a failure event, encompassing economic, environmental, and societal losses. In IMP, we include two reward models. The first is a *campaign cost* model where a global cost, $R_{camp}$, is incurred if at least one component is inspected or repaired, plus a surplus, $R_{ins} + R_{rep}$, per inspected/repaired component. This campaign cost explicitly incentivises agents to cooperate. The second is a *no campaign cost* model, where the campaign cost is set equal to 0 (i.e., $R_{camp} = 0$), and only component inspections and repairs costs are considered. Acting on finite-horizon episodes that span over $T$ time steps, all agents aim at maximising the expected sum of discounted rewards $\mathbb{E}[R_0] = \mathbb{E}\left[\sum_{t=0}^{T-1} \gamma^t \left[R_{t,f} + \sum_{a=1}^{n} \left(R_{t,ins}^a + R_{t,rep}^a\right) + R_{t,camp}\right]\right]$.

**Real-world data** While IMP policies are trained based on simulated data, they policies can then be deployed to applications where real-world streams of data are available. In that case, the damage condition of the components is updated based on collected real-world data, e.g., inspections.

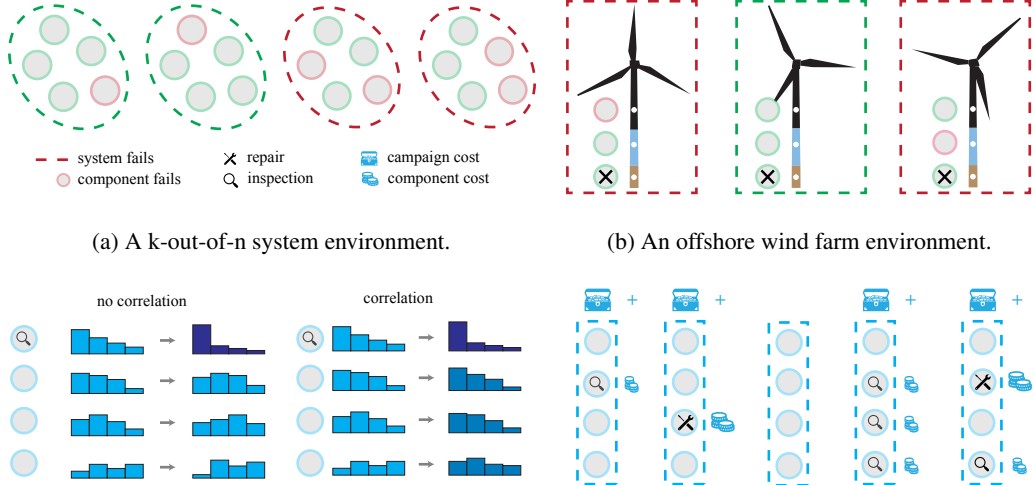

(a) A k-out-of-n system environment.

(b) An offshore wind farm environment.

(c) Uncorrelated and correlated initial damage distribution.

(d) A campaign cost environment.

Figure 2: Visual representation of available IMP-MARL environment sets and options. In 2a, a 4-out-of-5 system fails if 2 or more components fail. In 2b, a wind turbine fails if any constituent component fails. In 2c, when the environment is under deterioration correlation, the information collected by inspecting one component also influences uninspected components. In 2d campaign cost environments, a global cost is incurred if any component is inspected and/or repaired plus a surplus per inspected/repaired component.

## 3.3  IMP-MARL environments

With IMP-MARL, we provide three sets of environments to benchmark cooperative MARL methods. For all three, components are exposed to fatigue deterioration during a finite-horizon episode, inducing the growth of a crack over $T$ time steps. The first set of environments is named *k-out-of-n system* and refers to systems for which a system failure occurs if (n-k+1) components fail. Those systems have been widely studied in the reliability engineering community [40]. The second type of generic environment is named *correlated k-out-of-n system* and is a variation of the first one for which the initial components' damage distributions are correlated. The last one is named *offshore wind farm* and allows the definition of environments for which a group of offshore wind turbines must be maintained. The proposed IMP-MARL environment sets and options are graphically illustrated in Figure 2, and we hereafter provide details about these sets of environments. Additionally, the deterioration processes and implementation details are formally described in Appendices B and C.

**k-out-of-n system** In this set of environments, the components' damage probability distribution, $p(d_t^a)$, is defined as a vector of 30 bins, with each bin representing a crack size interval. Here, the failure probability of one component is defined as the probability indicated in the last bin. The specificity of a k-out-of-n system is that it fails if (n-k+1) components fail, establishing a direct link between the system failure probability and the component failure probabilities. For this first system, the initial damage distribution is statistically independent among components and the time horizon is $T = 30$ time steps. Since it is finite, we normalise each time step input and we define $s_t = (p(d_t^1), ..., p(d_t^n), t/T)$ and $o_t^a = (p(d_t^a), t/T)$. The interest of this system is that, in many practical scenarios, the reliability of an engineering system can be modelled as a *k-out-of-n system*.

**Correlated k-out-of-n system** The second set of environments is the same as the first one previously defined, with the difference that the initial damage distribution is correlated among all components. Therefore, inspecting one component also provides information about other uninspected components, depending on the specified degree of correlation. This setting is particularly challenging when approached from a decentralised scheme without providing components' correlation information to individual agents. To further address this issue, and in addition to their 30-bin local damage probability, the agents perceive correlation information $\alpha_t$ common to all, which is updated based on inspection outcomes collected from all components. We thus have: $s_t = (p(d_t^1), ..., p(d_t^n), \alpha_t, t/T)$

and $o_t^a = (p(d_t^a), \alpha_t, t/T)$. This damage correlation structure is inspired by practical engineering applications where initial defects among components are statistically correlated due to the fact that components undergo similar manufacturing processes [13].

**Offshore wind farm** The third set of environments is different from the previous ones as it considers a system with a certain number of wind turbines. Specifically, each wind turbine contains three representative components: (i) the top component located in the atmospheric zone, (ii) the middle component in the underwater zone, and (iii) the mudline component submerged under the seabed. In this case, we consider that the mudline component cannot be inspected nor repaired, as it is installed under the seabed in an inaccessible region and, since only the top and middle components can be inspected or repaired, two agents are assigned for each wind turbine. Furthermore, the damage probability, $p(d_t^a)$, is a vector with 60 bins and transitions differently depending on the component location in the wind turbine, as corrosion-induced effects accelerate deterioration in certain areas. Besides individual component damage models, inspection techniques and their associated costs also depend on the component location: it is cheaper to inspect or repair the top components than the middle one [41]. Moreover, while the mudline component cannot be directly maintained, its damage probability also impacts the failure risk of a wind turbine. In offshore wind farm environments, a wind turbine fails if any of its three constituent components fails, and the overall system failure risk is defined as the sum of all individual wind turbine failure risks. In this case, $p(d_t^a)$ is modelled as a 60-bin vector and the time horizon is $T = 20$. In this set of environments $s_t = (p(d_t^1), ..., p(d_t^n), t/T)$ and $o_t^a = (p(d_t^a), t/T)$.

**Implementation** All defined IMP environments are integrated with well-known MARL ecosystems, i.e., Gym [42], Gymnasium [43], PettingZoo [44] and PyMarl [9], through wrappers. The tested MARL methods are adopted from PyMarl's library, but other libraries are also compatible with our wrappers, e.g., RLlib [45], CleanRL [46], MARLlib [47], or TorchRL [48]. All developments are available on a public GitHub repository, `https://github.com/moratodpg/imp_marl`, featuring an open-source Apache v2 license.

# 4 Benchmark campaign of MARL methods

## 4.1 Tested methods

In an extensive benchmark campaign, we test seven RL methods: one fully centralised, one fully decentralised, and five CTDE approaches. The centralised controller, which has an action space that scales exponentially with the number of agents, is trained with the fully centralised method DQN [1] and is the only method taking $s_t$ as input. Furthermore, the fully decentralised method we test is IQL [20], in which all agents are independently trained. Regarding the five CTDE methods, we investigate three value-based methods: QMIX [16], QVMix [17], and QPLEX [18]. They factorise the value function during training, allowing agents to independently select actions during execution after a centralised value function is jointly learnt during training. The last two are the CTDE actor-critic methods COMA [19] and FACMAC [11]. They train independent policy networks and rely on a single centralised critic during training. Appendix D provides a detailed description of each method as well as a discussion of other methods of interest that are not included in the benchmark. We selected these methods for our benchmark study because they are well established and their implementations are open-sourced and available within the PyMarl framework [9].

All investigated MARL methods are compared against a representative baseline in the reliability engineering community [21, 36]. This baseline, referred to as expert-based heuristic policy, consists of a set of heuristic decision rules that are defined based on expert knowledge. The heuristic policy includes both parametric and non-parametric rules. Parametric decision rules depend on two parameters: (i) the inspection interval and (ii) the number of inspected components. On the other hand, non-parametric rules involve taking a repair action after detecting a crack and prioritising component inspections with higher failure probability. To determine the best heuristic policy, and for each environment, we evaluate all parametric rule combinations over 500 policy realisations, thereby identifying the heuristic policy that maximises the expected sum of discounted rewards among all policies evaluated.

Table 1: Number of agents specified in all investigated IMP environments.

| IMP environments | Number of agents | | | | |
|---|---|---|---|---|---|
| k-out-of-n system | 3 | 5 | 10 | 50 | 100 |
| Correlated k-out-of-n system | 3 | 5 | 10 | 50 | 100 |
| Offshore wind farm | 2 | 4 | 10 | 50 | 100 |

## 4.2 Experimental setup

The above-mentioned seven MARL methods are tested in the three sets of IMP environments defined in Section 3.3. The environments differ by the number of agents and by whether or not they include a campaign cost model. The numbers of agents tested in the six types of environments are presented in Table 1. To objectively interpret the variance associated with the examined MARL methods, 10 training realisations with different seeds are executed in each environment. As explained in Section 3, an agent makes decisions based on its local damage probability, the current normalised time step, and sometimes correlation information is additionally provided; while the state, used by DQN and CTDE methods, encompasses all of the information combined. In all cases, the action space features three possible discrete actions per agent, except for DQN, where the centralised controller selects an action among the $3^n$ possible combinations. For complexity reasons, we only test DQN in k-out-of-n environments featuring 3 and 5 components, as well as in environments with 1 and 2 wind turbines. Detailed information on rewards, observations, and states can be found in Appendix C.

Given the importance of hyperparameters on the performance of RL methods [49], we initially selected their values reported by the original authors. In an attempt to objectively compare the examined methods, parameters that play the same role across methods are equal. Notably, the learning rate and gamma, among others, are identical in all experiments. The controller agent network features the same architecture in all methods, consisting of a single GRU layer with a hidden state composed of 64 features encapsulated between fully-connected layers and three outputs, one per action, except for DQN, where the network output includes $3^n$ actions. In our case, DQN's architecture includes additional fully-connected layers and a larger size of hidden GRU states. Moreover, following common practice, agent networks are shared among agents, and thus a single agent network is trained. Specifically, we train only one network that is used for all agents, instead of training $n$ distinct agent networks. The training process with a single agent network improves data efficiency because the same episode can be used to perform $n$ backpropagations through the same agent network, using $n$ different observations. In contrast, only one backpropagation per agent network would be possible with a single episode if training is performed with $n$ different agent networks. To allow diversity in agents' behaviour, a one-hot encoded vector is also added to the input of this common network to indicate which one of the $n$ agents is making the decision. In CTDE methods, critics or mixers are also incorporated at the training stage with specific architectures according to each method and environment configuration. In most cases, the neural networks are updated after each played episode based on 64 episodes sampled from the replay buffer, which contains the latest 2,000 episodes. The only exception is COMA, which follows an on-policy approach, where the network parameters are updated every four episodes. For value-based methods, the training episodes are played following an epsilon greedy policy, whereas test episodes are executed with a greedy policy. The epsilon value is initially specified as 1 and linearly decreases to 0.05 after 5,000 time steps. This is different for COMA and FACMAC. Appendix E and the source code list more details and all parameters.

The number of time steps allocated for one training realisation is 2 million time steps for all methods. These 2 million training time steps are executed with training policies, e.g. epsilon greedy policy, saving the networks every 20,000 training time steps. To evaluate them, we execute 10,000 test episodes and obtain the average sum of discounted rewards per episode per saved network. These test episodes are executed with testing policies, e.g. greedy policy. We show in Appendix E.3 that 10,000 test episodes are needed due to the variance induced in the implemented environments. We emphasise that 10 training realisations are executed with different seeds for the same parameter values. All parameters are listed in Appendix E and in the source code.

# 5 Benchmark results and discussion

The results from the benchmark campaign are presented in Figure 3, showcasing the relative performance of MARL methods with respect to expert-based heuristic policies in terms of their expected sum of discounted rewards. In each boxplot of Figure 3, each of the 10 seeds is represented by its best policy, which achieved the highest average sum of discounted rewards during evaluation. We further explain the connection between learning curves and boxplots in Appendix F, Figure 10. Our analysis relies on relative performance metrics because the optimal policies are not available in the environments investigated. Additionally, the corresponding learning curves and the best-performing policy realisations can be found in Appendix F.

**MARL-based strategies outperform expert-based heuristic policies.** While heuristic policies provide reasonable infrastructure management planning policies, the majority of the tested MARL methods yield substantially higher expected sum of discounted rewards, yet the variance over identical MARL experiments is still sometimes significant. In environments with no campaign cost, the performance achieved by MARL methods with respect to the baseline differs in configurations with a high number of agents, as shown at the top of Figure 3. In contrast, MARL methods reach better relative results in environments with a high number of agents when the campaign cost model is adopted, as illustrated at the bottom of Figure 3. In general, the superiority of MARL methods with respect to expert-based heuristic policies is justified by the complexity of defining decision rules in high-dimensional multi-component engineering systems, where the sequence of optimal actions is very hard to predict based on engineering judgment [36].

**IMP challenges.** In correlated k-out-of-n IMP environments, the variance over identical MARL experiments is higher than in the uncorrelated ones, emphasising a specific IMP challenge. Under correlation, inspecting one component also provides information to uninspected components, impacting their damage probability and thus hindering cooperation between MARL agents. Another challenge is imposed in offshore wind farm environments, where the benefits achieved by MARL methods with respect to the baseline are also reduced in environments with a high number of agents. This can be explained by the fact that each wind turbine is controlled by two agents, being independent of other turbines in terms of rewards. Each agent must then cooperate closely with only one of all agents, hence complicating global cooperation in environments featuring an increasing number of agents.

**Campaign cost environments.** Yet another challenge can be observed in campaign cost environments under 50 agents, where MARL methods' superior performance with respect to heuristic policies is more limited. The aforementioned environments are challenging for MARL methods because agents should cooperate in order to group component inspection/repair actions together, saving global campaign costs. In addition, the heuristic policies are designed to automatically schedule group inspections, being favourable in this case. This is confirmed by the learning curves presented in Figures 11 and 12. On the other hand, in environments with more than 50 agents, MARL methods substantially outperform heuristic policies. At least one component is inspected or repaired at each time step and the results reflect that avoiding global annual campaign costs becomes less crucial.

**Centralised RL methods do not scale with the number of agents.** DQN reaches better results than heuristic policies, though achieving lower rewards than CTDE methods in most environments, despite benefiting from larger networks during execution. This highlights the scalability limitations of such centralised methods, mainly due to the fact that they select one action out of each possible combination of component actions.

**IMP demands cooperation among agents.** The results reveal that CTDE methods clearly outperform IQL in all tested environments, especially those with a high number of agents. This confirms that realistic infrastructure management planning problems demand coordination among component agents. Providing only independent local feedback to each IQL agent during training leads to a lack of coordination in cooperative environments, also shown by Rashid et al. [16]. However, the performance may be improved by enhancing networks' representation capabilities by including more neurons, yet this is true for all investigated methods.

**Infrastructure management planning via CTDE methods.** Overall, CTDE methods generate more effective IMP policies than the other investigated methods, demonstrating their capabilities for supporting decisions in real-world engineering scenarios. While Figure 3 presents the variance of the best results across runs, the learning curves further confirm this finding in Appendix F. In particular, QMIX and QVMIX generally learn effective policies with low variability over runs. Slightly more

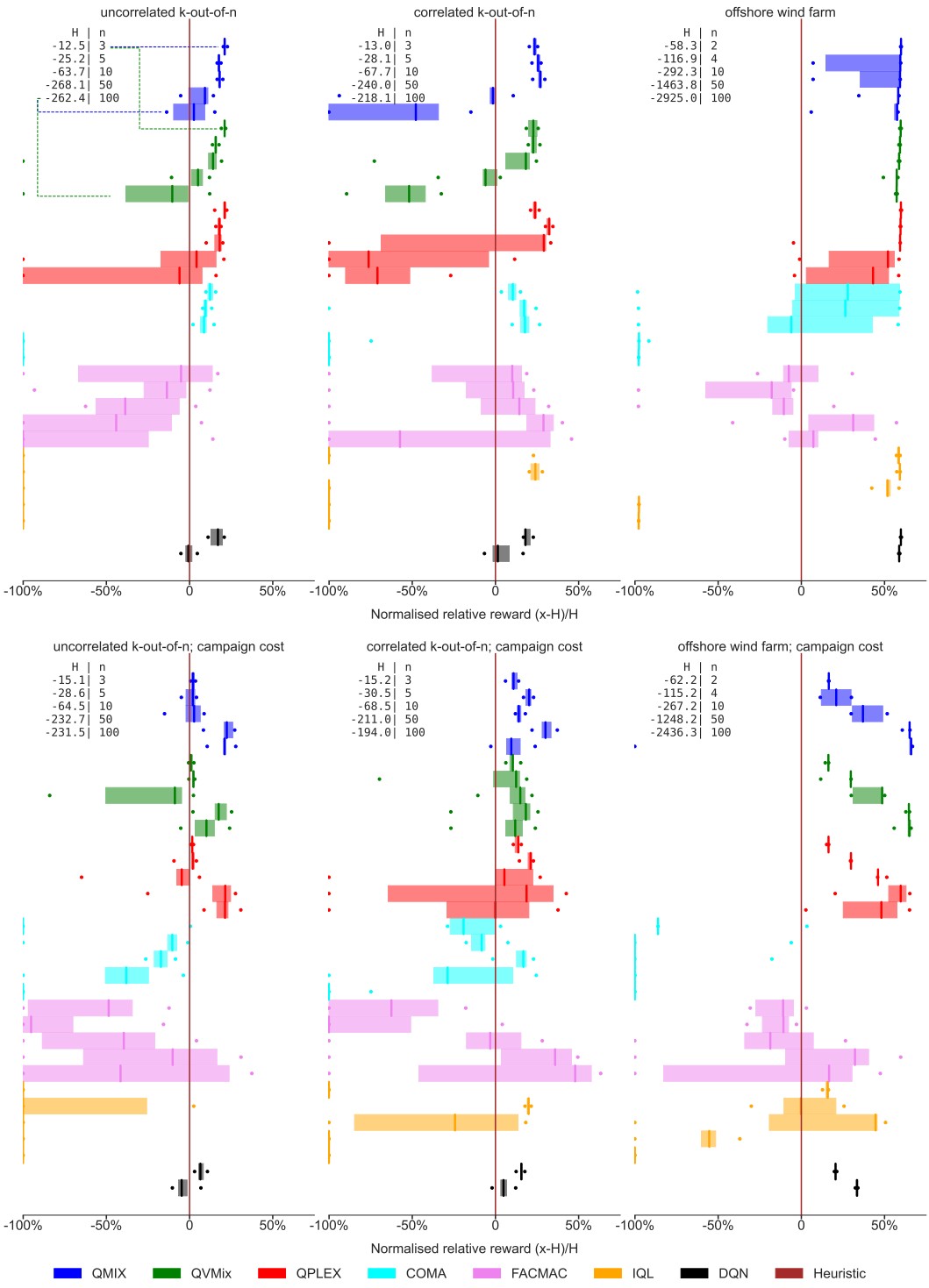

Figure 3: Performance reached by MARL methods in terms of normalised discounted rewards with respect to expert-based heuristic policies in all IMP environments, H referring to the heuristics result. Every boxplot gathers the best policies from each of 10 executed training realisations, indicating the 25th-75th percentile range, median, minimum, and maximum obtained results. The coloured boxplots are grouped per method, vertically arranging environments with an increasing number of $n$ agents, as indicated in the top-left legend boxes. Note that the results are clipped at -100%.

unstable, QPLEX also yields similar results to QMIX and QVMIX in terms of achieved results. While being able to outperform heuristic policies in almost every environment, FACMAC exhibits a high variance among runs. However, FACMAC effectively scales up with the number of agents and environment complexity (as reported in [11]), achieving some of the best results in IMP environments with over 50 agents as well as in correlated IMP environments. The results also suggest that COMA is the least scalable MARL method in our benchmark. This can be attributed to the fact that the computation of the critic's counterfactual becomes challenging with an increasing number of agents.

## 6   Conclusions

This work offers an open-source suite of environments for testing scalable cooperative multi-agent reinforcement learning methods toward the efficient generation of infrastructure management planning (IMP) policies. Through our publicly available code repository, we also encourage the implementation of additional IMP environments, e.g., bridges, transportation networks, pipelines, and other relevant engineering systems, whereby specific disciplinary challenges can be identified in a common simulation framework. Based on the reported benchmark results, we can conclude that centralised training with decentralised execution methods are able to generate very effective infrastructure management policies in real-world engineering scenarios. While the results reveal that MARL methods outperform expert-based heuristic policies, additional research efforts should still be devoted to the development of scalable cooperative MARL methods. While we model the IMP decision-making problem as a Dec-POMDP, modelling IMP problems as mean-field games [50] is a promising direction to be considered in environments with an increasing number of agents. Moreover, specific improvements are still required in environments where a global cost is triggered from the actions taken by any local agent, e.g., global campaign cost. Besides, more stable training is still needed in environments where local information perceived by one agent can influence the damage condition probabilities of others, as in the correlated IMP environments. In the future, more realistic and challenging environments for cooperative MARL methods could be investigated. One example would be assigning campaign costs to specific groups of components, instead of specifying only one global campaign cost.

## Acknowledgments and Disclosure of Funding

The authors acknowledge the computational resources provided by the Consortium des Équipements de Calcul Intensif (CÉCI), funded by the Fonds de la Recherche Scientifique de Belgique (F.R.S.-FNRS) under Grant No. 2.5020.11 and by the Walloon Region. We further acknowledge the insightful comments provided by our colleagues Adrien Bolland, Victor Dachet, Nandar Hlaing, Gaspard Lambrechts, Gilles Louppe, Matthias Pirlet, and Maurizio Vassallo.

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

# A    IMP-MARL public repository, license, data, and documentation

The new assets we provide in this paper are listed hereafter. IMP-MARL's suite of environments is publicly available on the GitHub repository `https://github.com/moratodpg/imp_marl`, featuring an open-source Apache v2 license. Moreover, IMP-MARL contains wrappers to facilitate its implementation on typical MARL ecosystems, i.e., Gym [42], RLLib [45], and PyMarl [9], as detailed in Section 4.

To reproduce the work reported in this paper, the following process can be executed: (i) cloning the repository, (ii) installing a virtual environment with the package requirements, and (iii) executing the script instructions of the corresponding method and IMP-MARL environments. In addition to the instructions for reproducing our results, we also provide tutorials to add new environments or wrappers.

We also provide the data files resulting from our experiments, enabling the reproduction of any reported result without re-running the experiments, as well as the corresponding implementation code, hence facilitating future cross-comparisons. The readily available results include Figures 3, 11, and 12, along with Tables 13, 14, 15.

Configuration, execution, and results files are permanently stored at Zenodo, accessible via `https://zenodo.org/record/8032339`. Additionally, the controller networks' weights of the best policies presented in Figure 3 are also stored there, thus fostering further interpretability studies of MARL-based strategies. The dataset is open-access and registered with the Digital Object Identifier (DOI) 10.5281/zenodo.8032339. More information and dedicated tutorials can be found on the repository.

# B    Modelling infrastructure management in IMP-MARL

In this appendix, we thoroughly describe the deterioration, inspection, and transition models implemented in this work. These models drive the dynamics of the IMP-MARL environments provided in this paper. Based on this information, one can easily learn how to create new environments.

## B.1    Deterioration models

The deterioration processes introduced here specifically correspond to fatigue deterioration mechanisms, yet corrosion, erosion, and many other practical infrastructure management problems can be similarly modelled.

**Correlated and uncorrelated k-out-of-n systems** Throughout the text and the code, the set of environments related to correlated and uncorrelated k-out-of-n systems are abbreviated as struct_c and struct_uc, respectively, or denoted as struct when referring to both of them. In the k-out-of-n environments currently included in IMP-MARL, the structural components are exposed to fatigue deterioration, and unless a repair is undertaken, the crack size $d_t$ (i.e., damage condition) evolves over time $t$ as [51]:

$$d_{t+1} = \left[\left(1 - \frac{m}{2}\right)C_{FM}S_R^m\pi^{m/2}n_S + d_t^{1-m/2}\right]^{2/(2-m)}, \tag{1}$$

where $\ln(C_{FM}) \sim \mathcal{N}(\mu = -35.2, \sigma = 0.5)$ and $m = 3.5$ stand for material variables, which directly influence the crack growth. Due to environmental and operational conditions, the components are subject to a dynamic load characterised by the stress range, $S_R \sim \mathcal{N}(\mu = 70, \sigma = 10 \text{ N/mm}^2)$, over $n_S = 10^6$ annual stress cycles, i.e., the number of load cycles experienced by the structural component in one year. At the initial step or after a component is repaired, the initial crack size is at its intact condition, defined by its initial distribution $d_0 \sim \text{Exp}(\mu = 1 \text{ mm})$, and a component level failure occurs when the crack size exceeds a critical size of $d_c = 20$ mm. The component failure probability $p_F$, defined as $p_F = p[g \leq 0]$, can be computed following a through-thickness failure criterion [52], where the failure limit at time step $t$ is formulated as $g_t = d_c - d_t$. At the system level, a failure event occurs if $k$ (out of $n$) components fail, and its corresponding system failure probability, $p_{F_{sys}}$, can be efficiently computed as a function of all components failure probabilities, as proposed in [40].

The continuous crack size is discretised into a certain number of discrete bins in order to enable efficient Bayesian inference once inspection indications are available. Further details can be found in

Table 2: Description of the discretisation scheme implemented.

| Environment | Variable | Interval boundaries | Bins |
|---|---|---|---|
| struct | $d_t$ | $[0, \exp\{\ln(10^{-4}) : (\ln(d_c) - \ln(10^{-4}))/28 : \ln(d_c)\}, \infty]$ | 30 |
| owf | $d_t$ | $[0, d_0 : (d_c - d_0)/(60 - 2) : d_c, \infty]$ | 60 |

Table 3: Variables specified in the offshore wind farm deterioration models.

| | Upper component | Middle component | Mudline component |
|---|---|---|---|
| $ln(C_{FM})$ | $\mu = -26.45$ $\sigma = 0.12$ | $\mu = -26.04$ $\sigma = 0.4$ | $\mu = -26.12$ $\sigma = 0.39$ |
| $m$ | 3 | 3 | 3 |
| $q$ | $\mu = 10.21$ $CoV = 25\%$ | $\mu = 7.40$ $CoV = 25\%$ | $\mu = 6.74$ $CoV = 25\%$ |
| $d_c$ | 20 | 60 | 60 |
| $n_S$ | 5,049,216 | 5,049,216 | 5,049,216 |

[13]. If the initial crack size among components is correlated (i.e., we are dealing with a correlated k-out-of-n system), the damage condition of each component is defined conditional on a common correlation factor, $\alpha$, via a Gaussian hierarchical structure [36]. In that case, the discretised damage bins should be defined conditional on the correlation factor. The specific discretisation implemented in our environments is defined in Table 2.

**Offshore wind farm** In this set of environments, a group of `n_comp` offshore wind substructures is considered, in which three representative structural components are modelled at different locations of the wind turbine: (i) at the atmospheric zone - upper level, (ii) at the splash zone - middle level, (iii) below the seabed - mudline. The deterioration, inspection, and cost models hence differ for each of the three considered components. While the fatigue deterioration is calculated according to Eq. 1, the expected dynamic load, $S_r$, is in this case defined based on industrial standards [53], as:

$$S_r = q\Gamma(1 + 1/\lambda)Y \,, \tag{2}$$

corresponding to the expected value of a Weibull distribution defined by the scale parameters listed in Table 3, $q \sim \mathcal{N}$, and shape factor, $\lambda = 0.8$, weighted by a geometric parameter, $Y \sim \mathcal{LN}(\mu = 0.1, \sigma = 0.1)$. The initial crack size distribution is specified for all wind turbine components as $d_0 \sim \text{Exp}(\mu = 0.11)$ and the remaining specific fatigue variables associated with each wind turbine component are listed in Table 3. At the wind turbine level, the failure event occurs if one component of the wind turbine fails. The wind turbine failure risk is then defined as the wind turbine failure probability multiplied by the consequences associated with a wind turbine failure event. At the wind farm level, the damage condition of a wind turbine does not influence the condition of the other wind turbines, and the wind farm system failure risk is defined as the sum of all turbines' failure risk.

## B.2 Inspection models

The inspection models implemented in IMP-MARL are hereafter described, defining the likelihood of retrieving a certain inspection outcome as a function of the damage size.

**Correlated and uncorrelated k-out-of-n systems** The inspection model is normally characterised depending on the accuracy of the measurement instrument, formally specified through probability of detection (PoD) curves, in which the probability of observing a crack is defined as a function of the crack size [13]. In this case, the inspection model is described by an exponential distribution $p(i_{d_t}|d_t) \sim \text{Exp}(\mu = 8)$, defining the probability of observing a crack during an inspection.

**Offshore wind farm** In this more practical set of environments, an eddy current inspection technique is here considered, whose PoD can be modelled according to industrial standards [53], as:

$$p(i_{d_t}|d_t) = 1 - \frac{1}{1 + (d_t/\chi)^b}, \tag{3}$$

where the factors $\chi$ and $b$ are specified as 0.4 and 1.43, respectively, for the upper component, but considered as 1.16 and 0.90, for the middle component. Naturally, less accurate inspection outcomes

can be expected for the middle component, as it is located in a region below the water level, where the visibility is reduced.

## B.3    Transition models

An overview of the transition model is explained hereafter. For a more detailed description, we refer the reader to [36]. Since the crack size is discretised in this work, the transition and inspection models can be stored in tables. In our code, they are encoded in Numpy files, which are stored in the repository folder `pomdp_models`. In particular, the files are named `Dr3031C10.npz`, `Dr3031_H08.npz`, and `owf6021.npz`, for k-out-of-n system, correlated k-out-of-n system, and offshore wind farm, respectively. By relying on already stored transition and inspection models, the environments can be simulated efficiently. Alternatively, the crack size evolution could also be directly computed at execution time, yet an additional computational expense would be then incurred.

The transition model can be defined based on the deterioration and inspection models previously described. If no inspection and maintenance are taken (i.e. do-nothing action), the damage condition progresses each time step according to the fatigue deterioration model formulated in Eq. 1. Note that in our three sets of environments, a time step represents a year. Considering that the damage follows a non-stationary deterioration process, the crack size distribution $d_{t+1}$ can be efficiently encoded as a function of the annual deterioration rate, $\tau_{t+1}$, and the crack size at the previous time step $d_t$ as $p(d_{t+1}|d_t, \tau_{t+1})$. Starting from $\tau_0 = 0$, the deterioration rate increases by one unit every year, unless a component is repaired, in which case the deterioration rate returns to the initial value. The deterioration evolution over a time step can be computed as:

$$p(d_{t+1}) = \sum_{\tau_{t+1}} \sum_{d_t} p(d_{t+1}|d_t, \tau_{t+1}) p(d_t) p(\tau_{t+1}) \,. \tag{4}$$

If an inspection action is planned, a damage indication $i_{d_{t+1}}$ is collected, and the crack size distribution can be updated via Bayes' rule:

$$p(d_{t+1}|i_{d_{t+1}}) \propto p(i_{d_{t+1}}|d_{t+1}) p(d_{t+1}) \,, \tag{5}$$

where the likelihood corresponds to the specific inspection model, described by a probability of detection curve, as mentioned before. Since the damage probabilities are discrete, the normalisation constant can be straightforwardly computed by simply summing the unnormalised bins [13].

To enable efficient computation of the deterioration evolution under correlation, a Gaussian hierarchical structure is adopted [36], in which the crack size probability is defined conditional on a common factor, $\alpha$ as $p(d_t|\alpha)$. In this work, we consider that the initial damage probabilities are equally correlated among components with a Pearson coefficient equal to 0.8.

The damage transitions, in this case, are formulated as:

$$p(d_{t+1}|\alpha) = \sum_{\tau_{t+1}} \sum_{d_t} p(d_{t+1}|d_t, \tau_{t+1}) p(d_t|\alpha) p(\tau_{t+1}) \,. \tag{6}$$

Once an inspection outcome is available, the common correlation factor is also updated based on the new information, thus influencing all components. The likelihood of collecting one inspection indication given $\alpha$ can be computed as:

$$p(i_{d_{t+1}}|\alpha) = \sum_{d_{t+1}} \left[ p(d_{t+1}|\alpha)\, p(i_{d_{t+1}}|d_{t+1}) \right] , \tag{7}$$

and the correlation factor can then be updated:

$$p(\alpha|i_{d_{t+1}}) \propto p(\alpha) p(i_{d_{t+1}}|\alpha). \tag{8}$$

Finally, the marginal damage probabilities are computed as:

$$p(d_{t+1}) = \sum_{\alpha} \left[ p(d_{t+1}|\alpha)\, p(\alpha) \right] . \tag{9}$$

Table 4: Main options available in IMP-MARL.

| Option name | Environment | Dict key | Type |
|---|---|---|---|
| Number of components | struct | `n_comp` | int |
| k components | struct | `k_comp` | int |
| Correlation | struct | `env_correlation` | bool |
| Campaign cost | struct | `campaign_cost` | bool |
| Number of wind turbines | owf | `n_comp` | int |
| Number of components per wind turbine | owf | `lev` | int |
| Campaign cost | owf | `campaign_cost` | bool |

Table 5: Observations options available in IMP-MARL.

| Option name | Env. | Dict key | Type | Dimensionality |
|---|---|---|---|---|
| Component damage probability | struct | *by default | float | 30 |
| Component deterioration rate | struct | `obs_d_rate` | float | 1 |
| All components damage probability | struct | `obs_multiple` | float | $\texttt{n\_comp} \cdot 30$ |
| All components deterioration rate | struct | `obs_all_d_rate` | float | `n_comp` |
| Correlation condition | struct | `obs_alphas` | float | 80 |
| Component damage condition | owf | *by default | float | 60 |
| Component deterioration rate | owf | `obs_d_rate` | float | 1 |
| All components damage condition | owf | `obs_multiple` | float | $\texttt{n\_comp} \cdot 60$ |
| All components deterioration rate | owf | `obs_all_d_rate` | float | `n_comp` |

## C    Options available in IMP-MARL and reward model

In IMP-MARL, the environments can be easily set up with specific options, from the definition of the number of agents to the observation information perceived by the agents and the state information received by mixers/critics. This can be straightforwardly specified through the configuration files provided on IMP-MARL's GitHub repository. These options are in fact parameters included in IMP-MARL's classes. In particular, Table 4 lists the main options available.

In addition to the main options previously mentioned, it is also possible to tailor the information encoded in the observations and states. One can choose which local information, $o_t^a$, the agents will receive, as well as the global information, $s_t$, available during training. Tables 5 and 6 list all possible options. Since these options are coded as booleans, the selected configuration can be easily defined by assigning a $True$ value. Additional details can be found in the code.

In the experiments conducted in this work, the selected parameters are listed in Table 7. Through the code provided in IMP-MARL, future works may investigate alternative observation and state information options.

### C.1    Reward model

The goal of the agents is to maximise the expected sum of discounted rewards, $\mathbb{E}[R_0] = \mathbb{E}\left[\sum_{t=0}^{T-1} \gamma^t \left[R_{t,f} + \sum_{a=1}^n \left(R_{t,ins}^a + R_{t,rep}^a\right) + R_{t,camp}\right]\right]$, as stated in Section 3.2. The rewards

Table 6: States options available in IMP-MARL.

| Option name | Environment | Dict key | Type | Dimensionality |
|---|---|---|---|---|
| All component damage condition | struct | `state_obs` | float | $\texttt{n\_comp} \cdot 30$ |
| All components deterioration rate | struct | `state_d_rate` | float | `n_comp` |
| Correlation condition | struct | `state_alphas` | float | 80 |
| All component damage condition | owf | `state_obs` | float | $\texttt{n\_comp} \cdot 60$ |
| All components deterioration rate | owf | `state_d_rate` | float | `n_comp` |

Table 7: Options set up in our experiments.

| Option name | struct_uc | struct_c | owf |
|---|---|---|---|
| state_obs | True | True | True |
| state_d_rate | True | True | False |
| state_alphas | False | True | False |
| obs_d_rate | False | False | False |
| obs_multiple | False | False | False |
| obs_all_d_rate | False | False | False |
| obs_alphas | False | True | False |
| env_correlation | False | True | False |
| campaign_cost | True & False | True & False | True & False |

Table 8: Rewards specified in our experiments.

| Component | Campaign cost | $R_{ins}$ | $R_{rep}$ | $c_f$ | $R_{camp}$ |
|---|---|---|---|---|---|
| struct | False | -1 | -20 | -10,000 | 0 |
| | True | -0.2 | -20 | -10,000 | -5 |
| owf upper level | False | -1 | -10 | -1,000 | 0 |
| | True | -0.2 | -10 | -1,000 | -5 |
| owf middle level | False | -4 | -30 | -1,000 | 0 |
| | True | -1 | -30 | -1,000 | -5 |

collected at each time step may include inspection $R_{ins}$ and repair $R_{rep}$ costs for all considered components, along with the system failure risk, which is defined as the system failure probability $p_{f_{sys}}$ multiplied by the associated consequences of a failure event $c_f$, formulated as $R_f = p_{f_{sys}} \cdot c_f$. Additionally, a campaign cost $R_{camp}$ may also be included if that option is active. The discount factor is defined as $\gamma = 0.95$ in our experiments and the specific rewards are listed in Table 8.

## D  Cooperative MARL methods

We considered methods from two families of algorithms in MARL: value-based and policy-based. Here, we propose a brief description of these methods, starting from single-agent RL (SARL) definitions to MARL. However, for more detailed information, we refer the reader to the original papers. Furthermore, in addition to the definition of Dec-POMDP in Section 3.1, we define the value function, also $V$ function, which evaluates the current joint policy $V^{\boldsymbol{\pi}}(s) = \mathbb{E}[R_t|s_t = s, \boldsymbol{\pi}]$. Another function of interest is the state-joint-action value function $Q_{tot} = Q^{\boldsymbol{\pi}}(s, \boldsymbol{u}) = \mathbb{E}[R_t|s_t = s, \boldsymbol{u_t} = \boldsymbol{u}]$, also called $Q$ function. The individual $Q$ function is defined by $Q^{\boldsymbol{\pi},a}(s, u) = \mathbb{E}[R_t|s_t = s, u_t^a = u, \boldsymbol{\pi}]$ or $Q_a$ for short. The reward is common to all agents and implies that $Q_{tot} = Q_a \forall a$. The advantage function is defined as $A(s, u) = Q(s, u) - V(s)$.

Value-based methods aim to learn the optimal $Q$ function defined as $Q^{\pi^*}(s, u) = \max_\pi Q^\pi(s, u)$. This enables the agent to greedily select the action $\pi^*(s) = \operatorname{argmax}_u Q^{\pi^*}(s, u)$. In SARL, this is accomplished through Q-learning [54], originally designed for tabular settings. However, as the size of the state-action space increases, it becomes impractical to compute $Q$ for each state-action pair. A solution, named DQN [1], approximates $Q$ with a neural network $\theta$ and learn $Q(s, u; \theta)$ by minimising the loss $\mathcal{L}(\theta) = \mathbb{E}_{\langle . \rangle \sim B}\left[\left(r_t + \gamma \max_{u \in \mathcal{U}} Q(s_{t+1}, u; \theta') - Q(s_t, u_t; \theta)\right)^2\right]$ where $B$ is a replay buffer composed of transitions $\langle s_t, u_t, r_t, s_{t+1}\rangle$ and $\theta'$ is the target network, a copy of $\theta$ updated periodically. This approach can train a centralised learner in a Dec-POMDP if all agents can access the state $s$ during execution, resulting in the learning of $Q_{tot} = Q^{\boldsymbol{\pi}}(s, \boldsymbol{u})$. Issues are that the joint action space scales exponentially with $n$, and in practice, agents select their action based only on their history of observation $(o, \tau)$ and not the state $s$. There is a decentralised solution, named IQL [20], which consists in learning independently $Q_a$. However, there are also CTDE methods that take the state $s$ into account during training. A solution proposed in QMIX [16] is to approximate $Q_{tot}$ as a function of all $Q_a$ and $s$ during training. Agents select actions based on their $Q_a$, which are now utility functions that factorise $Q_{tot}$ and not $Q$ function. One condition is that individual

$Q_a$ satisfy the individual global max (IGM): $\text{argmax}_{\boldsymbol{u_t}} Q(s_t, \mathbf{u_t}) = \bigcup_a \text{argmax}_{u_t^a} Q_a(\tau_t^a, u_t^a)$ [55]. In QMIX, the factorisation is achieved by a constrained hypernetwork [56] which links $s$ and $Q_a$ to $Q_{tot}$. QVMix [17] extends QMIX with the Deep Quality-Value method [57, 58], learning both $V$ and $Q$, using the former as the target of the latter. QPLEX [18] extends QMIX with the dueling structure $Q = V + A$ [59], learning a factorisation of $V$ and $A$ with transformers [60]. We selected these methods based on their IGM consistency, code availability, and results in the literature.

Policy-based methods learn directly the optimal policy through a neural network $\pi(s, u; \theta)$ that maximises $J(\theta) = \mathbb{E}_{\pi_\theta}[R_0]$. The well-known REINFORCE method [61] ascends the gradient $\nabla_\theta J = \mathbb{E}[\sum_t R_t \nabla_\theta \log \pi(u_t|s_t; \theta)]$ to find $\pi^*$. Actor-critic methods [62, 63] expand upon this method by incorporating a parameterised critic that estimates $Q(s_t, u_t; \phi)$, replacing $R_t$, with the actor serving as the parameterised policy. To reduce variance, a baseline $b(s)$ is injected into the gradient, usually $b(s) = V(s)$, and $Q(s, u; \phi)$ is replaced by $A(s, u; \phi)$ [64], leading to the new gradient expression $\nabla_\theta J = \mathbb{E}[\sum_t A(s_t, u_t; \phi) \nabla_\theta \log \pi(s_t, u_t; \theta)]$. Advantage estimation is accomplished either by $A(s_t, u_t; \phi) = Q(s_t, u_t; \phi) - \sum_u \pi(u|s_t; \theta) Q(s_t, u; \phi)$ or by $A(s_t, u_t; \phi) = r_t + \gamma V(s_{t+1}; \phi) - V(s_t; \phi)$. Extending these methods to MARL is a straightforward process and the decentralised solution is named IAC [19]. This approach involves each agent learning independently an actor and a critic, based only on the tuple $(\tau, o)$. However, this solution does not exploit the additional information provided by the state $s$. During training, the critic may exploit the full state $s$, which would result in a centralised critic. However, this approach provides the same feedback to all agents, missing out on the crucial aspect of credit assignment [65]. To address this, MADDPG [8] allows each agent to learn its own critic $Q_a(s, \boldsymbol{u}; \phi)$ that is considered centralised since its use of $s$ and $\boldsymbol{u}$. On the other hand, COMA [19] and FACMAC [11] propose solutions with a single centralised critic. FACMAC suggests using a central but factored critic by employing the value function factorization of QMIX. The joint-action $Q(s, \boldsymbol{u}; \phi)$ is built as a function of $Q_a(\tau, u^a)$, without the need to satisfy IGM and without the constraints on the hypernetwork. COMA, inspired by difference reward [66], proposes having the centralised critic compute a counterfactual baseline for each agent. For an agent $a$, the difference reward is $R(s_{t+1}, s_t, \boldsymbol{u_t}) - R(s_{t+1}, s_t, (\boldsymbol{u_t}^{-a}, c_t^a))$ where $c$ is a default action. Computing this requires simulating the environment steps several times. But in COMA, the centralised critic computes the advantage $A_a(s_t, \boldsymbol{u_t}; \phi) = Q(s_t, \boldsymbol{u}; \phi) - \sum_{u'^a} \pi(u'^a|\tau_t^a; \theta) Q(s_t, (\boldsymbol{u_t}^{-a}, u'^a); \phi)$ for each agent, allowing it to approximate $A$ without more environment steps. This has the cost of requiring to increase the input space of $A$ as it additionally takes $u^{-a}$ actions as input.

**Other methods** In addition to QMIX [16], QVMix [17] and QPLEX [18], QTRAN [55] and Weighted-QMIX [67] factorise $Q_{tot}$ differently from QMIX, but do not always satisfy IGM [55]. Other methods, such as MAVEN [68] and LAN [69], also extend over QMIX. The first improves exploration capabilities while the second learns to cooperate without factorising $Q_{tot}$. There are also policy-based methods that rely on the actor-critic paradigm, such as COMA [19] and FACMAC [11] with a single centralised critic. MADDPG [8] is another well-established method, which does not learn a single centralised critic, but one per agent and is designed for continuous action spaces. Another method, LIIR [70], aims to provide credit assessment with individual intrinsic rewards, while HATPRO and HAPPO [71] demonstrate that popular actor-critic methods like TRPO [72] and PPO [73] can be extended to cooperative MARL tasks. HATRPO and HAPPO could have been a great addition to our study but are unfortunately not implemented within the PyMarl library.

Another approach for dealing with cooperative multi-agent settings is to link the recent success of sequence models and reinforcement learning by using a multi-agent transformer (MAT) that learns to transform a sequence of observations into a sequence of actions, one per agent [74].

## E Experimental details

### E.1 Description of the parameters set up in the experiments

In this section, we provide the information required to reproduce the results reported in this paper. Since the neural networks are trained via MARL using PyMarl's [9] library, the parameters are here described following PyMarl's convention. However, their purpose can be easily deduced from the names themselves. The experimental parameters set up equal across all experiments are presented in Table 9, while the parameters specific to each method are hereafter detailed. Note that in Table 9, some parameters are not used by all methods, e.g., RMS parameters. Besides, target update intervals,

Table 9: Parameters set in our experiments.

| Parameter name | Parameters value | Parameter name | Parameters value |
|---|---|---|---|
| $\gamma$ | 0.95 | Time max | 2,050,000 |
| Target update | 200 | RMS epsilon | $10^{-5}$ |
| RMS alpha | 0.99 | Grad norm clip | 10 |
| Learning rate | 0.0005 | Obs last action | True |
| Agent network | [] - 64 GRU - [] | Save model interval | 20,000 |

Table 10: Exploration parameters.

| | Epsilon start | Epsilon finish | Epsilon anneal time |
|---|---|---|---|
| Value-base methods | 0.5 | .01 | 5000 |
| FACMAC | 0.3 | 0.005 | 50000 |

buffer size, and batch size are specified based on the number of episodes. When the number of agents increases, we only augment the number of trainable parameters of the mixer/critic networks, while the actor networks and other parameters are not modified. All the parameters can be found in the configuration files available on the GitHub repository and can be used to launch any of the experiments conducted in this paper.

Regarding the agent network representation for CTDE methods and IQL, it consists of one GRU layer with a hidden state that includes 64 features. This means that the input is fully connected to the 64 hidden states, which are then fully connected to the outputs, one per action. We represent it as "[] - 64 GRU - []". Since the number of actions, and hence the number of outputs, of the DQN network is $3^n$, a network with more representation capacity is needed. In that case, linear layers, whose number of output features are specified between brackets, are also included in the agent network surrounding the GRU layer. With $n = 2$ or $n = 3$ agents, the network is set as "[128] - 128 GRU - [128,64]" while for $n = 4$ or $n = 5$ agents, it is set as "[256] - 256 GRU - [256,256]". Note that the linear layers before the GRUs include a Relu activation function and the last taken action is also added to the observation of the agents as an additional input.

As mentioned previously, some parameters are common to almost all methods. For instance, the optimiser selected is RMSProp for all methods, except for FACMAC, which is trained with ADAM. In nearly all methods, the buffer stores the latest $2,000$ episodes, and at each episode, $64$ episodes are sampled to update the network. However, since COMA is an on-policy method, the networks are updated with the episodes just played. Therefore, in our experiments, COMA's networks are updated every four episodes based on these last experienced ones. This update is performed four times to ensure a fair amount of network updates with respect to the other methods, which are, in turn, updated every episode.

During training, the value-based methods rely on an epsilon-greedy policy, whose parameters are specified in Table 10, while they act following a greedy policy at testing. Note, however, that COMA and FACMAC utilise different training policies. FACMAC samples discrete actions through a Gumbel softmax for its actor, whereas exploration is performed via an epsilon, whose values are specified in Table 10. On the other hand, COMA follows a classic stochastic policy during training. At the testing stage, FACMAC and COMA select actions adhering to a greedy approach, selecting the action associated with the maximum probability.

In the conducted experiments, the parameters set up in the environments differ with respect to the number of agents and we distinguish three cases (i) $n <= 10$, (ii) $n = 50$, and (iii) $n = 100$. Starting with QMIX, the parameters are all the same when increasing $n$, except for the architecture of the mixing network, whose embedding size in the middle of the mixer is: (i) 32, (ii) 64, and (iii) 128. QMIX relies on a double-Q feature, i.e., the loss computed to update $\theta$ differ from the original. While in the original loss function, the target Q value used for the update is selected with the action that maximises the target Q value parameterised by $\theta'$, in double-Q, the action is the one that maximises the Q value parameterised by $\theta$. Therefore, we have: $\mathcal{L}(\theta) = \mathbb{E}_{\langle . \rangle \sim B}\big[\big(r_t + \gamma Q(s_{t+1}, u*; \theta') - Q(s_t, u_t; \theta)\big)^2\big]$ where $u* = \mathrm{argmax}_u Q(s_{t+1}, u; \theta)$. This target network is updated every 200

Table 11: Number of trainable parameters in uncorrelated k-out-of-n systems.

| Method | Network | $n = 3$ | $n = 5$ | $n = 10$ | $n = 50$ | $n = 100$ |
|--------|---------|---------|---------|----------|----------|-----------|
| QMIX | Agent | 27,587 | 27,715 | 28,035 | 30,595 | 33,795 |
|      | Mixer | 18,657 | 41,249 | 133,569 | 5,430,657 | 42,202,113 |
| QVMix | Agent | 27,587 | 27,715 | 28,035 | 30,595 | 33,795 |
|       | Mixer | 64,771 | 110,083 | 295,043 | 10,891,779 | 84,437,891 |
| QPLEX | Agent | 27,587 | 27,715 | 28,035 | 30,595 | 33,795 |
|       | Mixer | 58,249 | 83,289 | 155,269 | 1,830,933 | 4,901,797 |
| COMA | Agent | 27,587 | 27,715 | 28,035 | 30,595 | 33,795 |
|      | Critic | 35,971 | 45,955 | 70,915 | 1,195,907 | 10,840,323 |
| FACMAC | Agent | 27,587 | 27,715 | 28,035 | 30,595 | 33,795 |
|        | Critic | 48,002 | 72,706 | 134,466 | 1,042,370 | 3,313,218 |
| IQL | Agent | 27,587 | 27,715 | 28,035 | 30,595 | 33,795 |
|     | / | 0 | 0 | 0 | 0 | 0 |
| DQN | Agent | 157,979 | 758,003 | / | / | / |
|     | / | 0 | 0 | / | / | / |

episodes. QVMix is a variant of QMIX and the parameter values are similarly specified. In particular, QVMix contains two networks: (i) a Q network with the same architecture as QMIX, and (ii) a V network that is a copy of the Q network, but with only one output. As for QPLEX, we try to be close to the parameters selected for the SMAC experiments they conduct in their paper. When $n$ increases, we change only the attention layer size composed of $a$ layers and $b$ heads. In particular, we set up (i) $1L4H$, (ii) $2L4H$, and (iii) $2L10H$. The mixer embedding is $64$ for all experiments.

With respect to COMA, which is an actor-critic method, a few specific parameters should be additionally set up. The agent network, here denoted as the actor, is the same as for the other previously described. However, the critic varies with $n$ because the input of the critic becomes rather large, and its architecture is fully linear: (i) [128, 128], (ii) [512, 256, 128, 128], and (iii) [2048, 1024, 512, 256, 128]. A TD-$\lambda$ used to update the critic is set to $0.8$ and the learning rate to train the critic is the same as the one of the actor, specified in Table 9. In terms of FACMAC, the parameters of interest are those at the critic, as its size increases with the number of agents $n$. FACMAC features a 2-layer-mixing network and, therefore, we specify the size for both: (i) 64-64, (ii) 128-64, and (iii) 128-128. As in COMA, the TD-$\lambda$ parameter is set to $0.8$. The critic has a target network, similar to those included in value-based methods, that is also updated every 200 episodes with a soft target update with $\tau = 0.001$. Moreover, we follow the optimiser selection made in FACMAC's original paper and used ADAM, with an epsilon equal to $10^{-8}$.

Regarding IQL, which is a decentralised method, the parameters are identical in all tested environments. IQL also has the double-Q feature activated and learns independently individual Q values via DQN's algorithm. For DQN, we only run experiments with less than five agents, i.e., $n <= 5$. The main difference between tested environments is the size of their networks. Since DQN features $3^n$ outputs, only $64$ GRU cells are not enough in terms of network capacity. In our experiments, we increase the network size and confirm that DQN manages to achieve similar results as the other MARL methods. In DQN, the network architecture is the only parameter that is adjusted with the number of agents $n$.

Finally, we list in Table 11 the number of trainable parameters of the networks. The input is slightly different between the tested sets of IMP-MARL environments and, therefore, we show in the table only the parameters for one of them. Note that there is an agent network taking actions for all agents and we purposely duplicate the agent network row to emphasise that all agent's networks are identical across experiments.

## E.2 Hardware and experiments duration

Our experiments are all run on different clusters managed by SLURM [75]. They are executed with specific hardware requirements based on the number of agents: experiments with up to 10 agents are run on only CPUs, while we execute experiments on GPUs with 50 and 100 agents. The efficiency does not substantially improve when running experiments with less than 10 agents on

Table 12: Hardware configurations for training and testing experiments.

| Parameters | Train only on CPUs | Train on GPUs | Test only on CPUs | Test on GPUs |
|---|---|---|---|---|
| Number of CPU | 4 | 2 | 8 | 5 |
| RAM | 5 Gb | 6 Gb | 5Gb | 10 Gb |

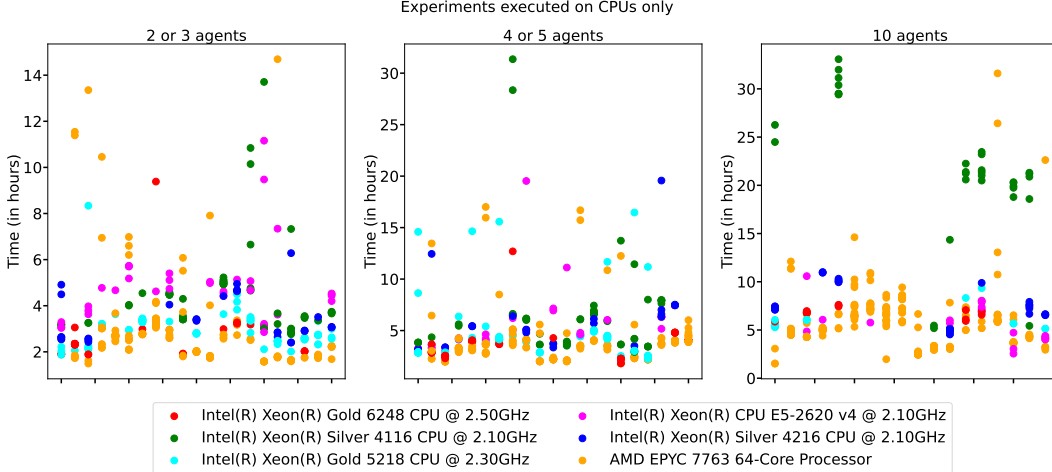

Figure 4: Training duration for experiments with $n <= 10$ performed on only CPUs.

GPUs because training a GRU layer requires forwarding the whole episode sequentially. In contrast, the computational time can be reduced when running experiments with 50 and 100 agents on GPUs because we train all agents as a single network and the batch size increases with $n$. We categorise the computational time required for the reported experiments according to whether (i) the experiment is (or is not) run only on CPUs, and (ii) the value reported corresponds to the training or the testing stage. In Table 12, we additionally provide the hardware requirements demanded during the training and testing phases. Note that we benefit from more resources during testing because 10 environments are running in parallel. Moreover, we intentionally demand more RAM to avoid problems. These reported RAM configurations are indicative and can be seen as requirements, yet not as exact memory usage numbers.

We represent the computational time required for the experiments during training in Figures 4 and 5 as well as during testing in Figures 6 and 7. To avoid overloading the figures, the markers do not explicitly indicate which experiment they correspond to, yet the experiments are all vertically grouped based on the method and the environment. For each abscissa, 20 experiments, with and without campaign cost, are represented. The first three sets of experiments represent QMIX in the uncorrelated k-out-of-n setting, followed by QMIX in the correlated k-out-of-n environment which is followed by QMIX in the offshore wind farm one. The methods are ordered as QMIX, QVMIX, QPLEX, COMA, FACMAC, IQL, and DQN, while the environments are ordered as k-out-of-n setting, correlated k-out-of-n, and offshore wind farm. It can be seen in Figures 4 and 6 that the two plots with $n < 10$ additionally represent the three additional experiments related to DQN.

The first observation that may be addressed is the resulting high variance. The variation across runs is logical because of the specific performance of the CPU/GPU models employed, but also due to the additional activity of clusters at the time of our experiments. With respect to the computational time required for training, testing episodes are not executed and we can see that, by running the experiments on only CPUs, we manage to train the agents in less than 10 hours, except for some occasional outliers. For experiments with 50 agents, and also relying on GPUs, the computational time is overall very similar to those previously mentioned, requiring less than 10 hours, yet a longer time is needed for those with 100 agents. The fastest training results correspond to COMA because we are running four environments in parallel, instead of one during training. We can see that IQL and QMIX follow closely, but QVMix, QPLEX, and FACMAC require additional computational time due to their architecture complexity. Naturally, the testing stage needs more time compared

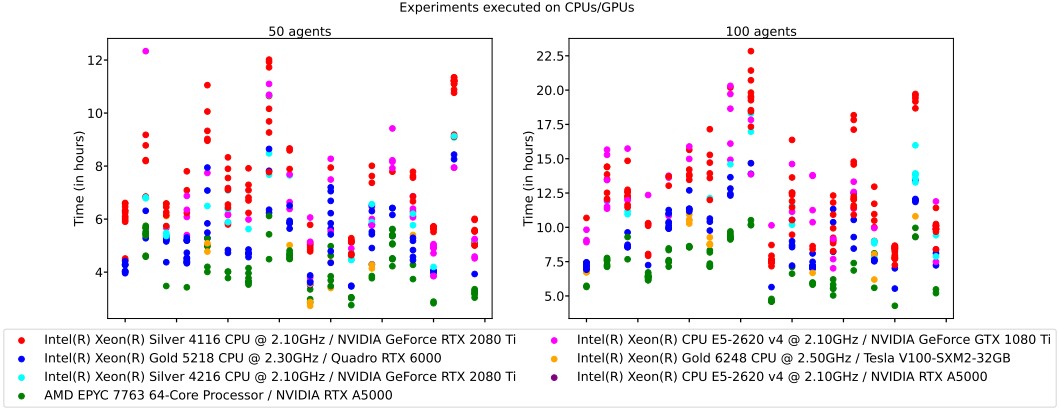

Figure 5: Training duration for experiments with $n >= 50$ performed on CPUs and GPUs.

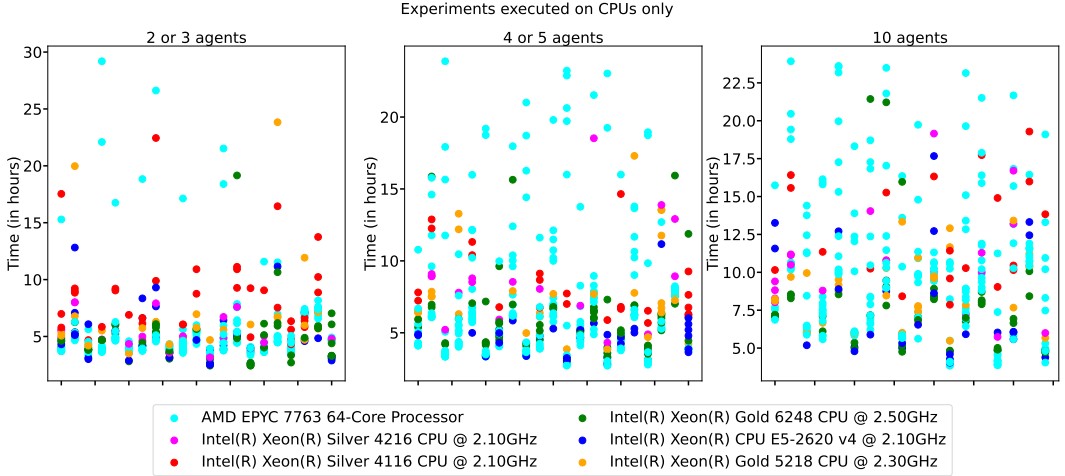

Figure 6: Testing duration for experiments with $n <= 10$ performed on only CPUs.

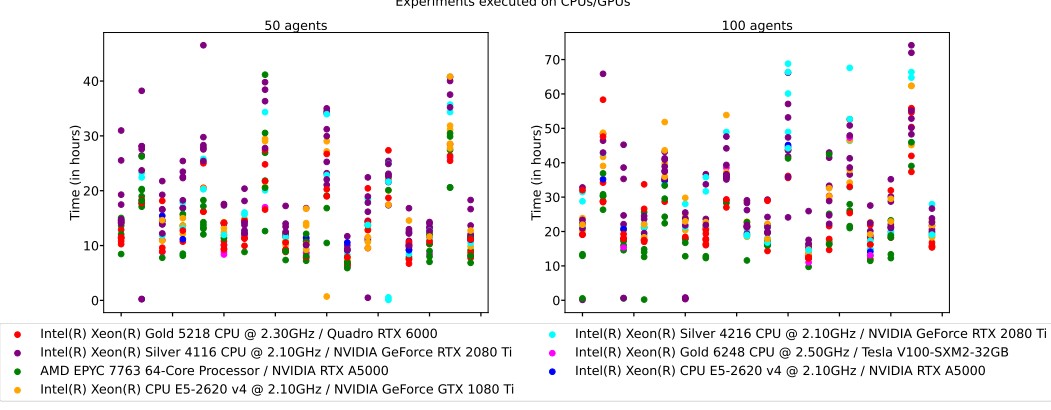

Figure 7: Testing duration for experiments with $n >= 50$ performed on CPUs and GPUs.

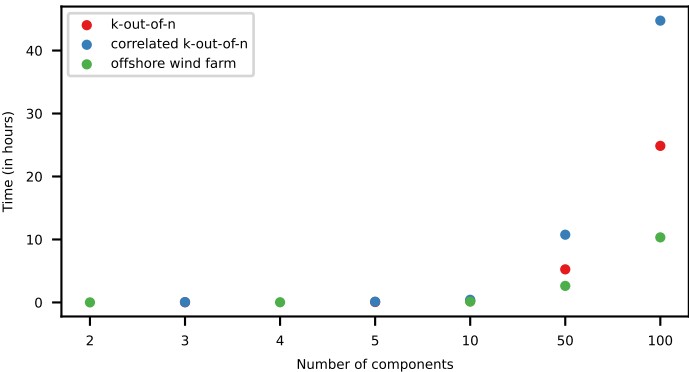

Figure 8: Computational time required for executing expert-based heuristic policies as a function of the number of components. The experiments are run on 2 AMD EPYC Rome 7542 CPUs @ 2.9GHz.

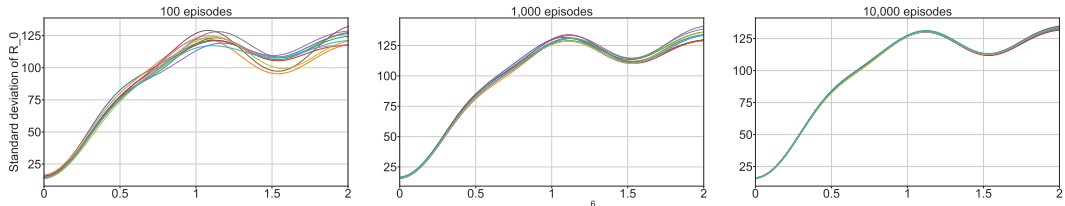

Figure 9: Variance analysis of a given set of neural networks. We report the standard deviation of the sum of discounted rewards obtained during the test phase. Each curve represents an entire test experiment when executing 100, 1,000, or 10,000 test episodes.

to training for all experiments because it executes $10,000$ test episodes per stored network. The correlated k-out-of-n environment slightly requires more time than the others because the correlation information is updated at every inspection step.

Furthermore, we represent in Figure 8 the time required for the computation of expert-based heuristic policies. The experiments are plotted as a function of the number of components and coloured based on their corresponding environment. In this case, all experiments are run on CPUs. We can see that heuristic policies can be efficiently computed for environments with less than 50 components, yet the computational time significantly increases for experiments with 50 or 100 components. This result is logical since the combination of evaluated parameters includes the number of components to be inspected at each inspection interval. Besides, the overall computation time is directly influenced by the time needed to run an episode, with the k-out-of-n environments taking longer compared to the offshore wind farm ones because the episode's finite horizon spans over 10 additional time steps.

### E.3 Statistical analysis of the variance associated with the number of test episodes

As previously explained, we conduct 10,000 test episodes to reduce the variance related to the expected sum of discounted rewards within a given environment. The choice is motivated by the direct relationship between the variance associated with $\mathbb{E}[R_0]$ and the number of test episodes. In our experiments, we average over 10,000 policy realisations, but here, we show that the standard deviation associated with $\mathbb{E}[R_0]$ for each trainig time step can vary significantly if an insufficient number of test episodes is simulated. Figure 9 illustrates the standard deviations observed when executing with 100, 1,000, and 10,000 test episodes. To produce this figure, we take the neural networks obtained with one FACMAC training run. These networks are trained over time in the offshore wind farm environment with 100 components. From this single set of networks, we execute 10 times the 100, 1,000, and 10,000 test episodes to observe how the variance evolves with this number of test episodes. We observe that the standard deviations of $\mathbb{E}[R_0]$ obtained with 100 test episodes significantly vary over the investigated test runs. Naturally, the variation of the standard deviation is reduced with an increasing number of test episodes, obtaining very similar standard deviations when testing with 10,000 episodes.

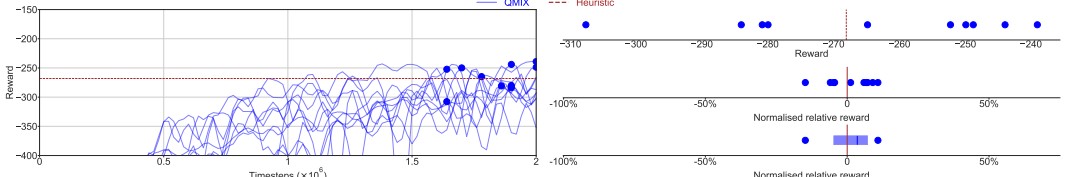

Figure 10: Visual description of the iterative process followed to generate the boxplots showcased in Figure 3. [Left] Learning curves corresponding to 10 QMIX training seeds in a k-out-of-n system with 50 agents. The markers highlight the policies that result in the highest expected sum of discounted rewards during evaluation, i.e., one policy per seed. [Right] The 10 selected policies are displayed at the top as a function of the expected sum of discounted rewards, along with the heuristic score obtained in this environment. In the middle plot, we calculate and represent the 10 selected policies as a function of normalised relative rewards, i.e., (x - h) / h. Finally, a boxplot is constructed at the bottom based on the previously calculated ten normalised relative rewards, each representing a different seed.

Moreover, one can also compute the largest difference that is observed between these 10 test runs at a specific time step i.e., the absolute difference of the estimated $\mathbb{E}[R_0]$ at a given time step. When testing 100 episodes, the absolute difference is 231.89 at a time step where the maximum of the 10 provided $\mathbb{E}[R_0]$ is $-2786.1$. If 1000 episodes are tested, the absolute difference is 99.8 where the maximum is $-2757.2$ at this time step, whereas by testing 10,000 episodes, the difference equals 24.8 with a maximum of $-2622.8$. These numbers represent only a trained set of networks over 1920 training runs, and thus a final conclusion cannot be claimed, yet it motivates the need of simulating 10,000 test episodes, as with only 100 test episodes, the absolute difference can reach up to 10 %.

# F   Additional benchmark results

In this section, we present additional results and remarks beyond those reported in the main text. In the first place, we provide the values of the expected sum of discounted rewards achieved by the best runs over all experiments. We list the best policy for each conducted experiment in Tables 13, 14, and 15. Note that the maximum values represented with markers at the right of each box in Figure 3 can be retrieved from these values by applying the normalisation (x-H)/H, with H being the value achieved by the heuristic policies. In Figure 10, we also visually illustrate the normalisation process that results in the boxplots presented in Figure 3.

Additionally, we represent in Figures 11 and 12 the learning curves corresponding to all our experiments. The learning curves showcase the evolution of the expected sum of discounted rewards every 20,000 training time steps, computed at the testing stage with 10,000 test episodes. Since the training is conducted with 10 different seeds for each environment and method, we also plot the corresponding 25th-75th percentiles around the median. These results confirm the variance observed between the best results and presented in Figure 3.

Based on Tables 13 and 14, one may additionally infer that correlated environments result in lower costs with respect to those uncorrelated. This is especially true for environments with n>=10 agents, specified without campaign costs, and in all environments set up with campaign costs. While MARL methods profit from the additionally provided correlation information, this is not always the case for the heuristic policies.

One final remark is that the discrepancy between the expert-based heuristic policy and MARL methods is more pronounced in offshore wind farm environments. This could be attributed to the shorter decision horizon or the higher cost per inspection in this particular case (see Table 15).

Table 13: k-out-of-n system best policies (* = campaign cost).

| n | QMIX | QVMix | QPLEX | COMA | FACMAC | IQL | DQN | Heuristics |
|---|------|-------|-------|------|--------|-----|-----|------------|
| 3 | -9.7 | -9.8 | -9.7 | -10.6 | -10.4 | -35.3 | -9.9 | -12.5 |
| 5 | -20.4 | -20.7 | -20.4 | -21.8 | -22.1 | -108.7 | -24.0 | -25.2 |
| 10 | -51.0 | -51.5 | -51.0 | -54.3 | -61.3 | -404.5 | / | -63.7 |
| 50 | -229.7 | -236.0 | -212.8 | -1190.6 | -249.0 | -1991.1 | / | -268.1 |
| 100 | -222.6 | -230.7 | -220.6 | -1770.1 | -225.7 | -1770.1 | / | -262.4 |
| *3 | -14.6 | -14.7 | -14.7 | -15.0 | -17.0 | -35.3 | -13.5 | -15.1 |
| *5 | -27.4 | -27.7 | -27.4 | -28.9 | -33.0 | -27.8 | -26.6 | -28.6 |
| *10 | -58.9 | -63.0 | -60.7 | -70.0 | -61.9 | -404.5 | / | -64.5 |
| *50 | -169.5 | -173.9 | -168.4 | -241.4 | -160.7 | -623.3 | / | -232.7 |
| *100 | -167.2 | -175.8 | -160.2 | -1770.1 | -144.8 | -1770.1 | / | -231.5 |

Table 14: Correlated k-out-of-n system best policies (* = campaign cost).

| n | QMIX | QVMix | QPLEX | COMA | FACMAC | IQL | DQN | Heuristics |
|---|------|-------|-------|------|--------|-----|-----|------------|
| 3 | -9.7 | -9.7 | -9.6 | -11.0 | -10.6 | -10.0 | -10.0 | -13.0 |
| 5 | -20.4 | -20.6 | -18.4 | -21.2 | -21.6 | -20.2 | -23.4 | -28.1 |
| 10 | -47.6 | -51.0 | -45.2 | -49.7 | -46.1 | -374.5 | / | -67.7 |
| 50 | -214.3 | -233.0 | -212.3 | -419.3 | -143.4 | -1339.9 | / | -240.0 |
| 100 | -250.3 | -289.0 | -276.8 | -486.9 | -118.3 | -1744.0 | / | -218.1 |
| *3 | -13.1 | -12.9 | -12.9 | -14.8 | -18.0 | -34.7 | -12.6 | -15.2 |
| *5 | -23.5 | -24.7 | -23.5 | -28.2 | -29.2 | -23.9 | -26.8 | -30.5 |
| *10 | -56.2 | -53.4 | -50.1 | -52.8 | -49.2 | -56.0 | / | -68.5 |
| *50 | -132.6 | -157.1 | -121.2 | -159.3 | -106.6 | -814.9 | / | -211.0 |
| *100 | -147.7 | -147.5 | -121.0 | -339.1 | -71.3 | -723.8 | / | -194.0 |

Table 15: Offshore wind farm best policies (* = campaign cost).

| n | QMIX | QVMix | QPLEX | COMA | FACMAC | IQL | DQN | Heuristics |
|---|------|-------|-------|------|--------|-----|-----|------------|
| 2 | -23.3 | -23.3 | -23.2 | -23.7 | -40.5 | -23.7 | -23.2 | -58.3 |
| 4 | -47.1 | -47.4 | -47.1 | -47.9 | -122.4 | -47.4 | -47.7 | -116.9 |
| 10 | -118.4 | -119.4 | -118.5 | -122.2 | -235.2 | -120.8 | / | -292.3 |
| 50 | -604.4 | -613.9 | -604.6 | -2805.8 | -627.3 | -2892.5 | / | -1463.8 |
| 100 | -1224.1 | -1238.8 | -1213.2 | -5785.1 | -1625.2 | -5785.1 | / | -2925.0 |
| *2 | -51.8 | -52.0 | -51.9 | -60.1 | -60.3 | -52.0 | -48.9 | -62.2 |
| *4 | -80.5 | -80.7 | -80.7 | -122.2 | -118.6 | -85.6 | -76.0 | -115.2 |
| *10 | -129.3 | -133.3 | -130.0 | -314.5 | -196.4 | -132.0 | / | -267.2 |
| *50 | -432.9 | -436.9 | -434.5 | -2892.5 | -502.8 | -1709.7 | / | -1248.2 |
| *100 | -808.1 | -829.0 | -852.3 | -5785.1 | -1280.5 | -5785.1 | / | -2436.3 |

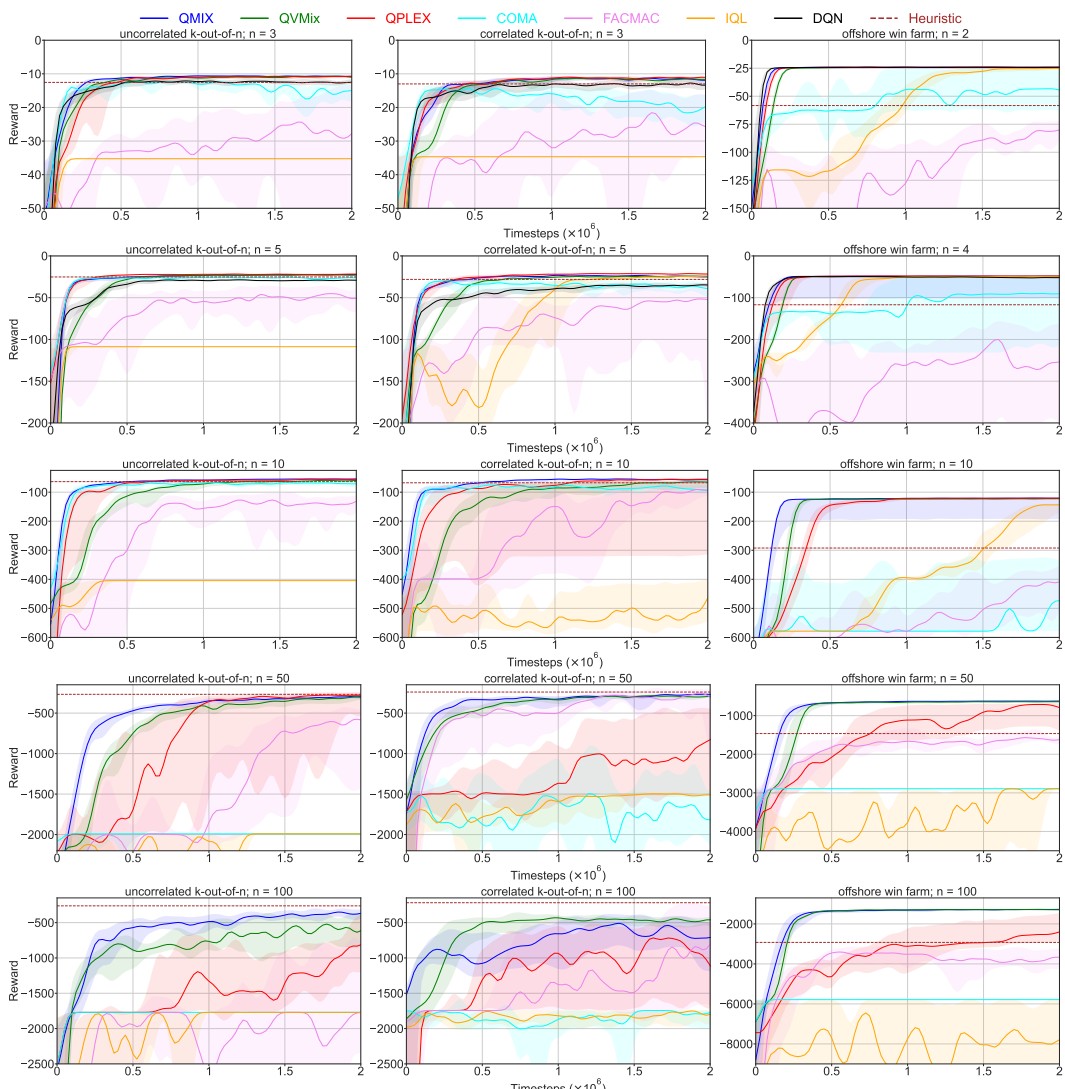

Figure 11: Learning curves in all environments with no campaign cost. Curves represent the sum of discounted rewards obtained during test time. The bold line is the median while the error bands are delimited by the 25th and 75th percentiles. Colours represent the different methods and the parameters of each environment can be inferred from the title above its corresponding graph.

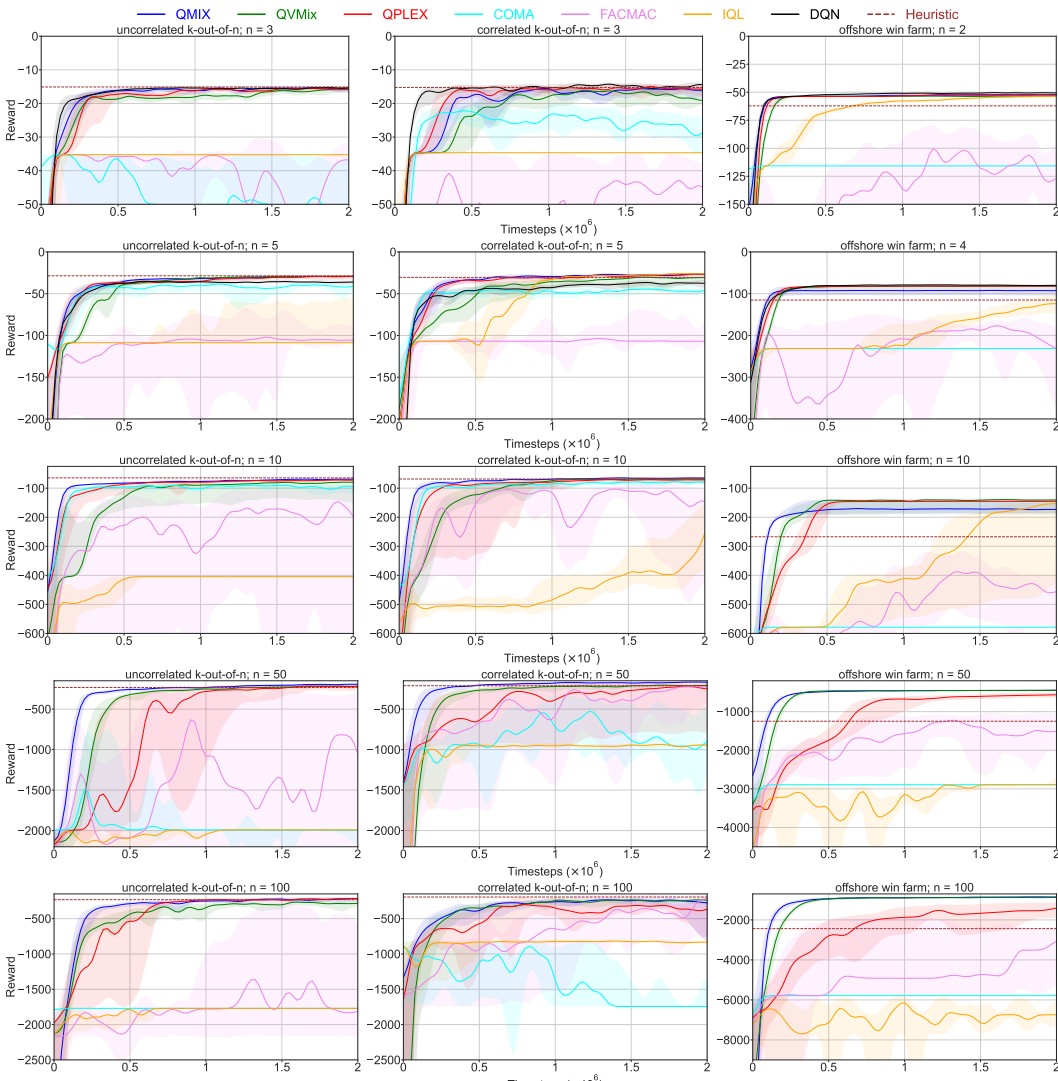

Figure 12: Learning curves in all environments with campaign cost. Curves represent the sum of discounted rewards obtained during test time. The bold line is the median while the error bands are delimited by the 25th and 75th percentiles. Colours represent the different methods and the parameters of each environment can be inferred from the title above its corresponding graph.

