# OpenReview forum: "IMP-MARL: a Suite of Environments for Large-scale Infrastructure Management Planning via MARL"
_NeurIPS.cc/2023/Track/Datasets_and_Benchmarks — NeurIPS 2023 Datasets and Benchmarks Poster_

### Official Review · Reviewer_FPDs · 2023-07-21
**Important and well-documented environments for benchmarking multiagent reinforcement learning on a real-world application**

**Rating:** 8
**Confidence:** 3

**Strengths:**

The paper is well written, mostly clear and the topic is timely, as no other similar benchmark of multiagent RL for infrastructure management planning exist.

The paper is thorough with several appendices detailing all parts of the paper.

I consider the contribution high, as predictive maintenance will be increasingly prevalent in industry and the importance of improving planning of inspections and service for large-scale manufacturing plants will increase. Hence, identifying promising methods will be important so to reduce the risk of performing costly real-world experiments on less promising methods.

I expect that this benchmarking suite could be relevant for the broader community focusing on planning industrial service operations.

The quality of the research is high.

**Additional Feedback:**

While Figure 1 is quite clear, I miss one or more figures that describe the architecture and the overall process flow of the execution of agents in the environments. I think this would improve the readability quite a bit.

It is not clear how decentralized agents share the agent networks, so I would suggest elaborating a bit on this aspect. I assume that not many sentences are needed to improve clarity.

Another aspect that is not totally clear from the text is the update of neural networks based on 64 episodes that are sampled from a replay buffer that contains the latest 2,000 episodes. Furthermore, it is hard to understand how the 10,000 test episodes are tested after 20,000 training time steps. I think this could be improved by a figure explaining the relationships between all these concepts.

In the caption of Figure 2 it is mentioned that “Every boxplot gathers the best policies from each of 10 executed training realisations …”. This is also mentioned elsewhere, such as “… presents the best results across runs …”. It is not clear to me what “best results” means here. Elsewhere in the text it is mentioned that variation is introduced by running the experiment 10 times with different seeds. Do not the results contain the performance of all 10 seeds? This is a bit confusing and could also be poor practice, see Henderson et al. (2018), unless I misunderstand something.

It is not clear to me what how the centralized methods differ in the action selection mentioned in line 307. I have a feeling that a figure, such as one of those mentioned above, could help explain this.

**Clarity:**

The paper is in generally well-written. See comments below for details on relatively minor improvements.

**Correctness:**

As far as I can understand the claims made in this paper are correct. The findings of the benchmark are discussed and compared with the state-of-the-art literature, and they seem to reproduce these results.

Both the evaluation methods and experiment designs seem appropriate to me.


**Documentation:**

Given the completeness of the documentation and code being open source, there should be enough details to reproduce the benchmarks.

Also, the Github page also provide several tutorials and guidelines that should help users to add new methods to the framework.


**Ethics:**

I see no ethical concerns.

**Limitations:**

Generally, I think there are few limitations besides from the obvious ones that could be the number of MARL methods and environments provided out-of-the-box. However, as this is not my area of expertise, I am not aware of any obvious omissions.

I only view the social impact as being positive and cannot see any negative effects, which is the view of the authors as well.

I am not sure I think the simulation of the offshore wind farms is done very well. Dividing the wind turbine into three different zones do not make much sense to me. It is not clear to me how one would do service to the middle part that is lowered into the water. Usually, moving and electric parts in the nacelle are the ones that are monitored for defects and maintained, as the non-moving parts are really hard to maintain, maybe with an exception for maintenance related to corrosion.

Another way the model seems too simplistic is that the progressions of the probability vectors utilize on physics-based models and not monitoring of real-time measurement. This can of course be improved upon, and I mean that this is not important for this paper but would be important for the applicability of models developed using this framework.

**Opportunities For Improvement:**

Some terms used in the paper are not well-defined and it forces quite a lot of content into a few pages. This reduces the clarity. Details are given below.

The significance of the paper is only related to multiagent reinforcement learning and large -scale infrastructure management planning, which could be considered too narrow. However, I mean that this would be a too narrow interpretation of its significance, as it is important to change the focus from games to real-world applications to increase the value of research on reinforcement learning.


**Relation To Prior Work:**

I find the relation to prior work to be described well.

**Summary And Contributions:**

The paper presents three environments for benchmarking multiagent reinforcement learning methods for infrastructure management planning. It also presents a benchmark of eight different RL methods for multiagent scenarios comprised of up to 100 agents that are compared to expert heuristics.

The contribution twofold: 1) the benchmarking environments are provided through open-source code and 2) the results of the benchmark using the default methods provided as part of the package as well.

---

> ### Author Response · Authors · 2023-08-14
> **Response to reviewer FPDs**
>
> Thank you for taking the time to review our paper and for sharing very informative comments.
> Your positive and supportive feedback is very much appreciated.
> In the revision of the manuscript, we have paid particular attention to clarifying potentially unclear statements.
> We hereafter carefully address your comments.
>
>
> > I am not sure .... related to corrosion.
>
> In the IMP-MARL environments implemented up to now, we have focused on structural components subject to fatigue and corrosion degradation. The fatigue deterioration process evolves more rapidly in certain areas (underwater), as documented by industrial standards. Underwater component inspections are usually conducted by experienced divers or remotely operated underwater vehicles. The cost of an underwater inspection is thus higher than that of a component located above the water. In the offshore wind-related environments, we then consider components in three distinct areas. Interestingly, the transition, inspection, and cost models are also specific to each considered location, making the decision-making problem more challenging at a system level. Designers and operators can easily adjust the implemented environments to identify infrastructure management planning strategies in case-specific scenarios.
>
> > Another way the .... using this framework.
>
> We also plan to include future environments with sensor installation actions and monitoring data collection.
> Monitoring information and physics models can still be combined through Bayesian updating if failure events related to structural components are considered.
> Additionally, we would also encourage the development of IMP-MARL environments for electrical/mechanical components, where failure statistics can be directly obtained from existing databases and/or real-world monitoring information.
>
> To encourage future developments in this direction, we have now incorporated suggestions in IMP-MARL's repository roadmap.
>
> > Some terms ....
>
> Due to space limitations, we recognise that specific cooperative multi-agent reinforcement learning concepts are not explained in detail or mainly discussed in the appendices. Based on your feedback, we have now provided further clarifications in the revised manuscript.
>
> > It is not clear .... to improve clarity.
>
> We have now revised the text in order to improve readability (**line 268**).
>
> > Another aspect .... training time steps.
>
> We also recognise that the redaction concerning the test/train/sampled episodes can be further clarified. The 64 sampled episodes over the latest 2000 episodes refer to specific off-policy settings. Off-policy MARL methods (all but COMA here) store the latest episodes in a replay buffer from which a certain number of episodes are uniformly sampled to compute one update of the network. This is detailed in the third paragraph of Appendix E.1 (**line 872**).
>
> We have now clarified this in the main text by splitting the paragraph that describes the training parameters and the description of our experiments' training/testing setup (**line 283**).
>
> > In the caption .... misunderstand something.
>
> First of all, what we consider as "best result" refers to the policy that achieves the highest expected sum of discounted rewards over one training realisation (**line 144**). One training realisation involves executing 2 million training time steps and saving the policies (neural network parameters) every 20,000 timesteps. For each experiment/realisation (i.e. 1 seed), we test each of the 1,000 saved policies in terms of the expected sum of discounted rewards. Over one training realisation, the best policy hence corresponds to the saved policy that achieves the highest sum of discounted rewards (over the 1,000 stored ones).
>
> With respect to Figure 2, we executed 10 training realisations for every experiment, each of them assigned with a different seed. The best policy corresponding to each of these 10 training realisations is identified and represented in the boxplots showcased in Figure 2, denoted as "the best results across runs" (**line 340**).
>
> If you still think it is necessary, we could add a clarification with respect to this matter into the manuscript.
>
> > It is not clear to .... in line 307.
>
> As mentioned in the manuscript (**lines 224 \& 256**) , a centralised controller has access to all information ($s_t$) and selects one over 3 actions for each of the $n$ components of the environment, leading to a choice among the joint action space of $3^n$ possible actions.The main difference is that the centralised controller needs to select one action over the $3^n$ possible actions, instead of having $n$ decentralised controllers that select one action among three.
>
> We have also noted your suggestion of including an additional clarifying figure.
> Benefiting from the available additional space, we are definitely considering the incorporation of a figure aimed at helping the reader to better understand how agents are trained.

---

> > ### Comment · Reviewer_FPDs · 2023-08-27
> >
> > Thank you for your thorough response to my questions. It is much appreciated. I will take it and the revised paper into account when completing the review.

---

> > > ### Author Response · Authors · 2023-08-31
> > >
> > > Thank you very much for taking the time to respond to our comments and for the positive feedback.

---

### Official Review · Reviewer_y9Yt · 2023-07-21
**Important area for benchmarks but the paper falls short**

**Rating:** 4
**Confidence:** 4
**Correctness:** Yes.
**Clarity:** Yes.

**Strengths:**

RL and online planning in sequential decision-making hold significant promise in real-world applications. Yet, benchmark environments are often toy settings that simplify several real-world assumptions. As a result, this set of benchmark environments is an important step in establishing a suit of large-scale real-world domains where cooperative RL can be particularly useful. The paper is well-written and motivated, but in my opinion, the breadth of coverage of problems is not enough for this track at this moment. I would encourage the authors to include more problems and resubmit.

**Additional Feedback:**

N/A

**Documentation:**

Yes.

**Limitations:**

See above.

**Opportunities For Improvement:**

There are several other applications, e.g., robot warehousing, supply chain logistics, etc., that can particularly benefit from cooperative RL, with several papers in the past years from top multi-agent conferences. My only concern is that the suit of problems presented in this paper is not large enough to establish a benchmark setting just yet. The authors should include more environments in the set.

**Relation To Prior Work:**

Yes

**Summary And Contributions:**

The paper presents a set of benchmark environments related to large-scale infrastructure management for benchmarking cooperative reinforcement learning algorithms. There are several algorithms that the authors use for the benchmark problems and present results.

---

> ### Author Response · Authors · 2023-08-14
> **Response to reviewer y9Yt**
>
> Thank you for taking the time to review our paper and for sharing very informative comments.
> Your feedback is very much appreciated. Considering your positive comments with respect to the strengths of our contributions, we hope that with the response and applied modifications, your support can be further strengthened.
>
> From your feedback, we have found two potential interpretations regarding the possible extension of IMP-MARL's suite of environments.
>
> 1. Incorporation of more diverse infrastructure management planning environments: bridges, mechanical components, etc.
> 2. Incorporation of additional environments representative of real-world applications other than infrastructure management planning, such as robot warehousing and supply chain logistics, in order to enable more general MARL benchmarking studies.
>
>
> We address the first point (1) in the general response to all reviewers, where we concur on the importance of increasing the diversity of infrastructure management planning (IMP) environments. One of the core objectives proposed in this work is indeed enabling the creation and addition of new IMP environments/models. Finding open-source models or environments in the reliability engineering community is usually very hard. As stated in the manuscript, one of our main contributions is the release of IMP-MARL's open-source repository to encourage people to publish their IMP models, facilitated by the provided tutorials and examples.
>
> With respect to the second point (2), we acknowledge the significant impact that a collection of real-world scenarios would have on the MARL community. We think that our paper marks an additional step in MARL's engagement towards real-world problems. However, assembling a comprehensive assortment of real-world applications as a singular contribution is challenging to encapsulate in a 10-page paper, especially if it involves the implementation of new types of real-world environments, as the definition of each problem requires significant domain knowledge. This is particularly true when considering the additional information required in the Appendix of this paper to define the IMP problems accurately.
> Our primary aim remains the advancement of MARL for IMP problems.
> While this aligns with the machine learning community's pursuit of real-world experimentation over traditional games, it also urges the reliability engineering community to embrace RL as a solution for their challenges. Besides, IMP applications already pose interesting and specific challenges to the reinforcement learning community, as reported in the manuscript.
>
> Based on your comments, we have revised the related work section, mentioning an additional real-world application (**line 105**). If you have any further suggestions, we would be pleased to cite other relevant publications dealing with real-world applications.

---

### Official Review · Reviewer_3EU2 · 2023-07-24
**Interesting work with rigorous benchmarking for MARL in an important application domain**

**Rating:** 8
**Confidence:** 4
**Correctness:** Yes, the evaluation method and experi…

**Strengths:**

First, extensive benchmarking is done of the environments on several MARL algorithms. These are also compared with expert-based heuristic policies to analyze scenarios when learned policies fail. This shows the benefit of benchmarking exercise on new environments to highlight further avenues for RL research.
Second, the environments are released with Gym and PyMARL wrappers which allows easy use with RL packages. This is essential to facilitate fast adoption by the RL community.
Third, it is a good evidence of real-world utility of MARL (for reliability engineering) which is critical as we seek to move beyond game play.

I think this a strong paper which deserves to be accepted.

**Additional Feedback:**

Not applicable

**Clarity:**

The paper is well written and easy to follow. Also, the github repo is well documents. I enjoyed reading this work.

**Documentation:**

Yes, the code is well documented and paper is clearly written.

**Ethics:**

No ethical concerns.

**Limitations:**

Yes, the authors have mentioned limitations in discussion and conclusions.

**Opportunities For Improvement:**

Couple of directions:
1.Benchmark recent SOTA MARL algorithms such as HAPPO/HATRPO [1] and MAT [2]. These specifically excel with heterogeneous agents, have shown sota performance on recent MARL benchmarks and would be cool to see how they work in IMP environments.
2. Integrate real-world data streams in the environments. The environments present real case-studies in reliability engineering but calibrating the environment to real-data streams (for eg: specific wind farms in the world) can make this an even more attractive benchmark for RL researchers and bring them closer to real world impact.
3. Add more diverse environments. The three in the current suite, while very useful, have somewhat similar design. Considering net-zero goals and specific carbon management scenarios can make good future benchmark environments. These problems are very relevant to this benchmark, essential in the current need to focus on climate change and also give a good benchmark to test impact of MARL.

**Relation To Prior Work:**

Yes, this is clearly discussed. More details on other kinds of environments which they have not considered but would be interesting for future work in allied domains would be interesting.
This can also be a good starting point for contributors to this project to get started.

**Summary And Contributions:**

The work introduces IMP-MARL, a suite of cooperative MARL environments for large-scale infrastructure management planning. The environments can be configured to range from 2 to 100 agents; and are released with Gym and PyMARL wrappers to streamline utility. Performance is benchmarked by comparing policies learned using several MARL algorithms and also with expert-based heuristic policies. The work has interesting implications to advance both MARL and reliability engineering research.

---

> ### Author Response · Authors · 2023-08-14
> **Response to reviewer 3EU2**
>
> Thank you for taking the time to review our paper and for sharing very informative comments.
> Your positive and supportive feedback is very much appreciated. We hereafter carefully address your comments.
>
> > 1. Benchmark recent SOTA MARL algorithms such as HAPPO/HATRPO [1] and MAT [2] .... would be cool to see how they work in IMP environments.
>
>
> Despite our interest in HATRPO/HAPPO, our decision not to include them is mentioned in Appendix D (**line 841**) and justified in the main text (**line 234**). We have not tested them in the benchmark because they are not coded within the same implementation framework (i.e., PyMARL) as the other tested MARL methods. Even if the goal of this paper is not to identify the best of all MARL methods but to verify their effectiveness in generating effective IMP strategies, we decided not to compare methods implemented with a different framework in order to not bias the benchmark results [1]. We, however, motivate the creation and/or testing of other MARL methods, such as HATRPO/HAPPO, in future works.
>
> To facilitate this task, we have implemented additional wrappers, such as a wrapper for MARLLib's framework, where more MARL methods are included together with those based on PyMARL (please see the [pull request](https://github.com/moratodpg/imp_marl/pull/60) on our repo for further information). Moreover, we provide indications and information with respect to [the wrappers in IMP-MARL's repository](https://github.com/moratodpg/imp_marl/tree/main/imp_wrappers), allowing users to adapt IMP-MARL environments and test them following other frameworks. On top of that, we have also included a [roadmap](https://github.com/moratodpg/imp_marl/blob/main/ROADMAP.md) in the GitHub repository to highlight our future directions, also in terms of RL frameworks.
>
> With respect to MAT, we believe you refer to Multi-agent Transformers [2].
> Defining the IMP problem as a sequence model is indeed a great suggestion.
> We have now incorporated MAT in the paragraph of Appendix D related to other methods (**line 842**), and we have updated the reference to this appendix in the main text (**line 233**).
>
> [1] Gorsane, R., Mahjoub, O., de Kock, R. J., Dubb, R., Singh, S., \& Pretorius, A. (2022). Towards a standardised performance evaluation protocol for cooperative marl. Advances in Neural Information Processing Systems, 35, 5510-5521.
> [2] Wen, M., Kuba, J., Lin, R., Zhang, W., Wen, Y., Wang, J., \& Yang, Y. (2022). Multi-agent reinforcement learning is a sequence modeling problem. Advances in Neural Information Processing Systems, 35, 16509-16521.
>
>
> > 2. Integrate real-world data streams in the environments .... bring them closer to real world impact.
>
>
> Integrating real-world data is a great suggestion. While IMP policies are trained based on simulated data, the policies can be then deployed to applications where real-world streams of data are available. Within the implemented IMP-MARL environments, please note that inspection actions dictate the collection of information to update the knowledge of the components/system condition at a discrete point in time. Besides inspections, sensor installation actions can also be easily included as another continuous information collection action within the implemented IMP template environments.
> We have now revised Section 3.2, where the environment formulation is described, and have added a paragraph on this matter (**line 167**).
>
>
> Moreover, damage statistics collected from operating mechanical components (e.g., gearbox, generator) can be used directly in the environments’ transition model. Up to now, the focus of the environments is on structural components, where failure statistics cannot be directly obtained, and instead, the damage is computed via physics-based simulators.
>
> We, however, encourage future environments related to mechanical components. This future development has now been added to the [roadmap](https://github.com/moratodpg/imp_marl/blob/main/ROADMAP.md).
>
>
> > 3. Add more diverse environments ... on climate change and also give a good benchmark to test impact of MARL.
>
>
> We have addressed the topic regarding the diversity of the implemented environments in the general response to all reviewers.  As described in the general response, we definitely agree and encourage future developments in this direction. IMP-MARL has been specifically designed to facilitate the creation and implementation of new environments. To this end, we have created tutorials to assist users and researchers to add and test their models through IMP-MARL.

---

> > ### Comment · Reviewer_3EU2 · 2023-08-22
> > **Thanks for your response**
> >
> > Thanks for your response and clarifications. I'll further update my rating.
> >
> > Good luck with this work.

---

> > > ### Author Response · Authors · 2023-08-31
> > >
> > > Thank you very much for taking the time to respond to our comments and for the encouraging words.

---

### Official Review · Reviewer_quZJ · 2023-07-25
**Social value of the infrastructure management**

**Rating:** 9
**Confidence:** 4
**Correctness:** The claims made in the submission are…
**Clarity:** The paper is succinct and well-written.

**Strengths:**

The reinforcement learning framework emphasizes the cooperation among different components in the complicated system. The cooperation reflects the complex network structure in the large-scale infrastructure system.

**Additional Feedback:**

The author may take into account the social valued of the infrastructure operation in the model. A successful management of the infrastructure system also contributes to the well-being of the whole society.

**Documentation:**

The paper is well-organized and provides the related materials.

**Ethics:**

I have no concern about ethics of this paper.

**Limitations:**

The connection between this paper to the mean-field game is obvious. The authors may want to emphasize this connection in the paper.

**Opportunities For Improvement:**

There is a large literature of applied mathematics on the reliability analysis. The authors could connect the reinforcement learning algorithm to the computation difficulty of the reliability analysis.

**Relation To Prior Work:**

The authors clearly state their contributions relative to the literature in the paper.

**Summary And Contributions:**

The paper investigates a traditional problem by using reinforcement learning. The authors transform a reliability problem in a complicated system into a multi-agent reinforcement learning problem. This enlarges the application of the mean-field game, which connects applied mathematics and engineering.

---

> ### Author Response · Authors · 2023-08-14
> **Response to reviewer quZJ**
>
> Thank you for taking the time to review our paper and for sharing very informative comments.
> Your positive and supportive feedback is very much appreciated. We hereafter carefully address your comments.
>
> > The authors could connect the reinforcement learning algorithm to the computation difficulty of the reliability analysis.
>
>
> We agree that the reliability assessment of complex engineering systems poses a mathematical and computational challenge by itself. When dealing with multi-component engineering systems, the reliability could be efficiently approximated via surrogate models, and agents trained through reinforcement learning could then decide if additional information from high-fidelity simulations and/or sensors should be additionally collected. Although this development is currently out of the scope of this paper, we will certainly consider this in future implementations.
>
>
> > The connection between this paper to the mean-field game is obvious. The authors may want to emphasise this connection in the paper.
>
> The extension toward mean-field games, which allows modelling the environment with an infinite number of agents, is indeed a very promising direction when considering environments with an increasing number of components.
> We have added a new sentence to encourage the investigation of mean-field games in future works (**line 359**). We would be open to any suggestion you may have in terms of other references that you think would be interesting to cite within the context of mean-field games.
>
> > The author may take into account the social valued of the infrastructure operation in the model. A successful management of the infrastructure system also contributes to the well-being of the whole society.
>
>
> The societal significance of identifying effective infrastructure management planning strategies is clear. We have addressed this comment in the general response and further mentioned it in the revised manuscript.

---

### Author Response · Authors · 2023-08-14
**General response**

We would like to thank all reviewers for their insightful comments, which will contribute to strengthening the quality of the manuscript.

We address here two general comments, regarding the societal impact and diversity of the implemented environments, both mentioned by more than one reviewer.

Specific comments are directly provided to each reviewer separately. We quote the original comments posed by the reviewers before responding to them. Note that the quoted text is sometimes reduced to ensure the individual responses comply with space limitations.

Note that the revised version of the manuscript has been uploaded with modifications formatted in red colour. In our responses, we refer to the line numbers in the revised version.

The societal impact created by generating effective infrastructure management planning policies has been pointed out by reviewers **quZJ**, **3EU2**, and **FPDs**. We further emphasise this in the revised version, providing stronger arguments on the importance of this work in the introduction (**lines 54-60**) as well as in the list of contributions (**line 78**).

With respect to the implemented IMP-MARL environments, reviewers **3EU2**, **FPDs**, and **y9Yt** suggest enhancing the diversity of the implemented IMP-MARL environments.

Encouraging diversity lies at the core of our work's objective. As mentioned in the related work section, most implementations and codes are not open-source in publications reported by the reliability engineering community. As a result, it is very challenging to implement and investigate a diverse set of environments. In addition to showcasing the effectiveness of MARL algorithms in generating policies for IMP applications, our work aims to initiate an open-source movement within (but not limited to) the infrastructure engineering community. This is why we emphasise that our IMP-MARL suite has been specifically designed to facilitate creating and implementing new environments. As pointed out by **FPDs**, we have created tutorials to assist users and researchers to add and test their models through IMP-MARL.

---

### Decision · Program_Chairs · 2023-09-22

**Decision:**

Accept (Poster)

**Comment:**

3 of the four reviewers were generally quite supportive of this benchmark based on infrastructure planning (IMP), and recommended acceptance.

One reviewer (y9Yt) does bring up some valid criticisms--especially in terms of the kinds of assumptions underlying the benchmark.

In accordance with the majority opinion, I will recommend acceptance.

Having looked at the reviews, the discussion and the  paper, I was sympathetic to the concerns of y9Yt. In particular, I think more attention needs to be paid to the connection to the planning/MDP literature. The International Planning Competition--especially the stochastic track--had looked at infrastructure planning like benchmarks (including such old chestnuts as the SysAdmin problem). It would be useful to put benchmark baselines from that model-based inference literature too--instead of just going with simple heuristic approaches. I say this in part because there was at least one case--during IPC 2021/ICAPS 2021--when a competition benchmark-- The Flatland Challenge -- designed specifically showcase MARL approaches for a class of inference problems wound up being won by model-based approaches. See this link for details:  https://icaps21.icaps-conference.org/Competitions/